# Computational inference of cancer-specific vulnerabilities in clinical samples

Kiwon Jang[1†], Min Ji Park[2†], Jae Soon Park[1], Haeun Hwangbo[1], Min Kyung Sung[1], Sinae Kim[2], Jaeyun Jung[2], Jong Won Lee[3], Sei-Hyun Ahn[3], Suhwan Chang[2*] and Jung Kyoon Choi[1,4*]

* Correspondence: suhwan.chang@amc.seoul.kr; jungkyoon@kaist.ac.kr

[†]Kiwon Jang and Min Ji Park contributed equally to this work.
[2]Department of Biomedical Sciences, University of Ulsan College of Medicine, Asan Medical Center, Seoul 05505, Republic of Korea
[1]Department of Bio and Brain Engineering, KAIST, Daejeon 34141, Republic of Korea
Full list of author information is available at the end of the article

## Abstract

**Background:** Systematic in vitro loss-of-function screens provide valuable resources that can facilitate the discovery of drugs targeting cancer vulnerabilities.

**Results:** We develop a deep learning-based method to predict tumor-specific vulnerabilities in patient samples by leveraging a wealth of in vitro screening data. Acquired dependencies of tumors are inferred in cases in which one allele is disrupted by inactivating mutations or in association with oncogenic mutations. Nucleocytoplasmic transport by Ran GTPase is identified as a common vulnerability in Her2-positive breast cancers. Vulnerability to loss of Ku70/80 is predicted for tumors that are defective in homologous recombination and rely on nonhomologous end joining for DNA repair. Our experimental validation for Ran, Ku70/80, and a proteasome subunit using patient-derived cells shows that they can be targeted specifically in particular tumors that are predicted to be dependent on them.

**Conclusion:** This approach can be applied to facilitate the development of precision therapeutic targets for different tumors.

## Introduction

There have been substantial efforts to profile cancer dependency by loss-of-function screens in cell lines [1–10], providing valuable resources that can lay foundation for new ways to fight cancer. However, the screening methods are applicable only to in vitro cell culture, limiting the discovery of therapeutic targets for clinical samples. A patient-derived cell model cannot be generated in a sizeable fraction of patients. In other patients, the time needed to develop the model is too long for clinical decision-making. Furthermore, in the absence of matched normal samples, it is difficult to identify truly cancer-specific dependencies and characterize them in association with somatic alterations.

Gene suppression will affect cell survival through the perturbation of the gene regulatory network. Perturbed transcriptomes, not basal transcriptomes, are directly linked to the consequential phenotypes. However, perturbed transcriptomes associated with

cell death cannot be acquired experimentally because the expression patterns of only surviving cells will be captured. In addition, experimental data may be too complicated to capture the core changes that actually contribute to growth phenotypes because of biological complexity involving feedbacks and secondary effects. Practically, it is close to impossible to generate transcriptome data for knockdown of individual genes in each sample. Therefore, we need to simulate transcriptomic perturbations and link the resulting expression patterns to the state of cell death or growth.

To simulate all downstream events following gene suppression, we need a gene regulatory network represented as a directed graph encompassing all genes. In our previous work [11], we designed a comprehensive Bayesian prior model based on data for transcription factor binding, chromatin accessibility, enhancer-promoter interactions, and genetic expression association mapping. We then derived causal relationships from ~ 1400 breast cancer transcriptomes. Another option is to assign orientations to links in the coexpression network, such as ARACNe (Algorithm for the Reconstruction of Accurate Cellular Networks) [12], that represents direct regulatory interactions.

In this work, we sought to develop a computational method that predicts cancer-specific dependencies for clinical samples with breast cancer as a model. We used the results of genome-wide shRNA [3] and CRISPR-Cas9 [9, 10] screening of breast cancer cell lines, the breast cancer regulatory networks described above, and the transcriptomic and mutational profiles of TCGA breast cancer samples [13]. Systematic characterization of common dependencies and experimental validation using patient-derived cells identified novel therapeutic targets in breast cancer. This translational approach lays the foundation for future applications to discovering personalized therapeutic targets from clinical molecular data on various tumor types.

## Results

### Development of in silico CRISPR/RNAi and prediction model

The workflow of our method is illustrated in Fig. 1. Based on the gene regulatory network, we devised a method to simulate the perturbation effects of CRISPR or RNAi as detailed in the "Materials and methods" section. We employed the Bayesian network in breast cancer and the same type of network in liver cancer for a negative control from our previous works [11, 14]. In this work, we constructed the ARACNe network [12] for breast and liver cancer. We determined the orientation of the links to simulate perturbation effects.

We merged two independent CRISPR-Cas9 screens of 28 and 25 breast cancer cell lines [9, 10], each based on a dependency score named CERES and BAGEL [9, 15], respectively. We selected three cutoffs (CERES = − 0.6, − 1.0, or − 1.5 and BAGEL = 0, 2, or 4) to result in the similar number of dependencies per cell line between the two measures. As a result, the selected cutoffs led to 120,485, 44,644, and 3644 cases, respectively. From shRNA dropout screens of 77 breast cancer cell lines [3], each gene was assigned a normalized GARP (Gene Activity Ranking Profile) score, or zGARP, in each cell line [3, 16]. Here, we used three cutoffs (zGARP = − 2, − 3, or − 4) to define dependencies, resulting in 88,837, 31,229, and 12,412 cases, respectively. We randomly chose the same number of instances below (for BAGEL) or above (for CERES and zGARP) the given cutoffs as independency cases. For each dependency and

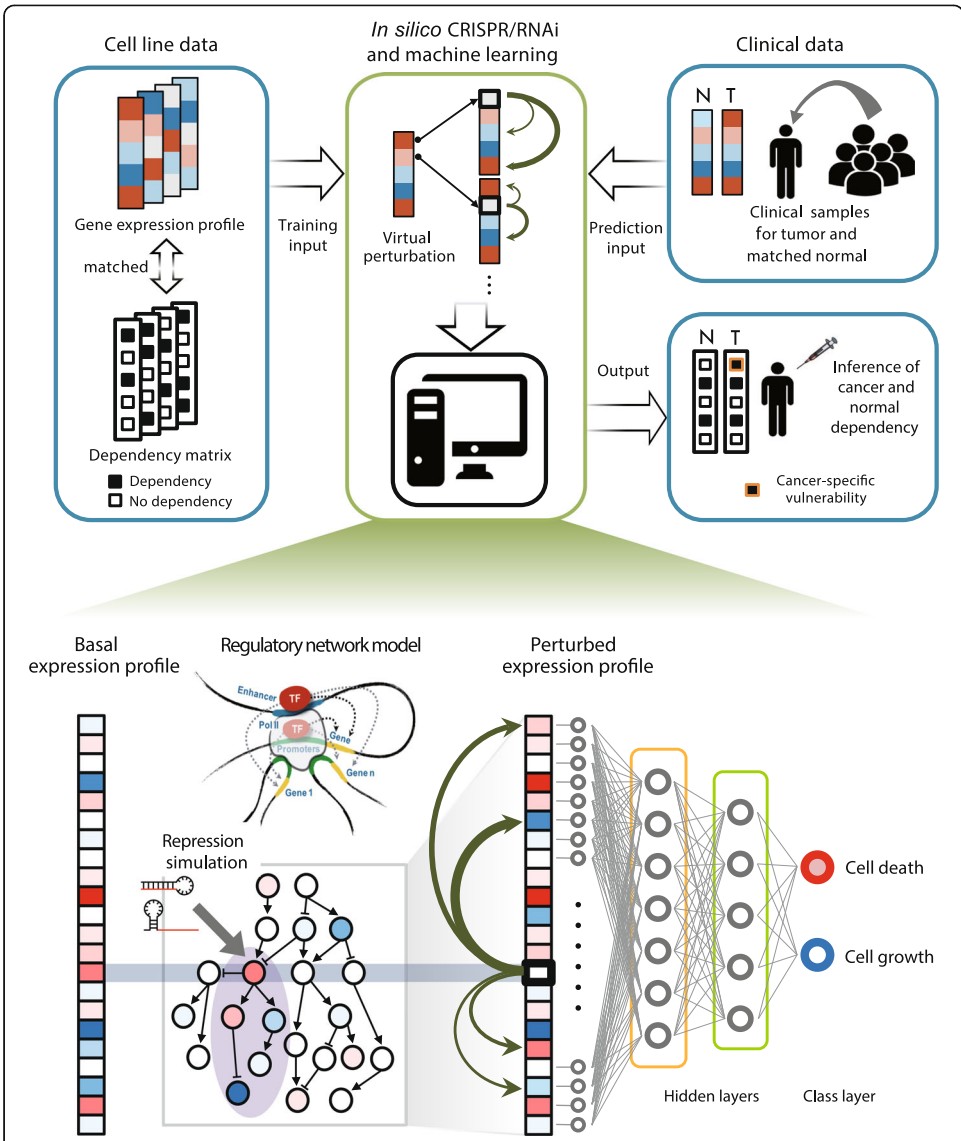

**Fig. 1** Schematic workflow for predicting cancer-specific vulnerability in clinical samples. Transcriptomes and matching dependency maps constructed for cancer cell lines are used for training. Tumor and matched normal transcriptomes of clinical samples are used as input for prediction. Cancer-specific vulnerability can be identified by comparing the prediction outcomes for tumor and normal samples. The prediction model consists of in silico CRISPR/RNAi and machine learning. For a given transcription profile derived from a cell line or clinical sample, virtual repression can be performed by adjusting the expression level of each target gene and its downstream genes. The perturbed expression profiles are fed into neural networks for training or prediction. The output of the model is the probability that the perturbed transcriptome is associated with cell death or that the survival of the given sample is dependent on the inactivated gene (see the "Materials and methods" section)

independency, we performed our in silico CRISPR/RNAi to generate the perturbed transcriptome.

We trained deep neural networks (DNN) with various hyperparameters (Additional file 2: Table S1) by using the perturbed expression patterns for the experimentally determined dependencies and independencies (Fig. 1). The ROC and precision-recall curves show that better performance can be achieved with more stringent cutoffs for both CRISPR and RNAi with the Bayesian network (Fig. 2a and Additional file 1:

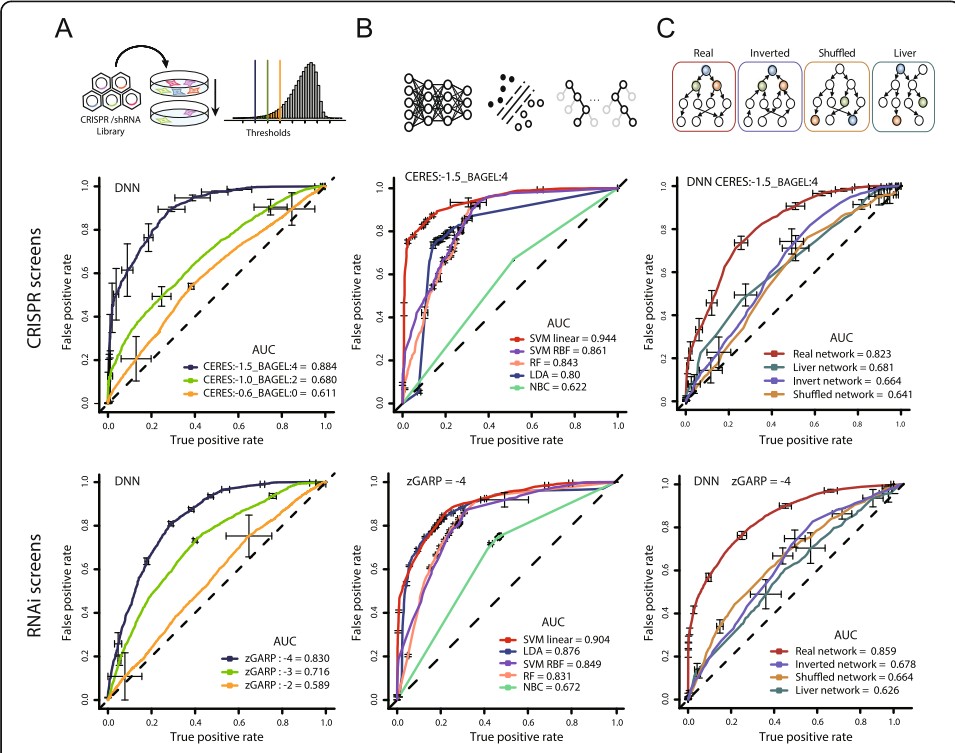

**Fig. 2** Prediction performance and validation of network-based simulations. **a** ROC curves from our DNN model for the CRISPR model (upper) and RNAi model (lower) according to different cutoffs of the cell line screen data. **b** ROC curves derived from different prediction models when the most stringent cutoff (zGARP = − 4, CERES = − 1.5, and BAGLE = 4) was used for training. **c** ROC curves from the DNN model with different test networks. We chose genes that have > 100 downstream genes in the regulatory network and used their perturbed expression patterns as training input. As negative controls, the breast cancer network was randomized by shuffling the nodes while maintaining the network structure (shuffled network) or inverted by reversing the orientation of all links (inverted network). In addition, the same types of networks were constructed by using liver cancer data (liver network). Shown are the averages of the five best models

Figure S1A). These results imply that true dependencies share more distinct patterns in their perturbed transcriptomes. Similar performance results were obtained when we repeated the model training using the ARACNe network (Additional file 1: Figure S2A).

We also tested other machine learning methods including support vector machine (SVM) with different kernels, naïve Bayes classifier (NBC), linear discriminant analysis (LDA), and random forest (RF). When the most stringent cutoff was used, all methods except NBC showed acceptable performance in terms of AUC and AUPR with both loss-of-function screens and both regulatory networks (Fig. 2b, Additional file 1: Figures S1B, and S2). We tested the linear, polynomial, sigmoidal, and radial basis function (RBF) kernels for SVM. The polynomial and sigmoidal kernels presented performance levels similar to or lower than those of radial SVM.

Most importantly, we tested the validity of our network-based simulation. To this end, we chose genes that have > 100 downstream genes in the network and used their perturbed expression patterns as training input. With these genes only, similar levels of prediction performance were achieved for both CRISPR and RNAi screens (red ROC curves in Fig. 2c). Three types of negative controls were prepared. First, we performed network randomization by shuffling the nodes while maintaining the network structure. Second, we generated inverted networks by reversing the orientations of all links.

Third, we constructed networks in liver cancer to model biologically irrelevant regulatory interactions. Accurate predictions were not made in these cases (see green, violet, and yellow ROC curves in Fig. 2c, Additional file 1: Figures S1C, and S2C). In addition, the real network resulted in a significantly better agreement with experimental expression perturbation data in MCF7 (Additional file 1: Figure S3).

## Characterization of prediction outcomes for clinical samples

We performed systematic in silico CRISPR/RNAi for 113 tumor and matched normal samples from TCGA and subjected the resulting expression profiles to our prediction models. First, we compared the TCGA prediction outcomes between the CRISPR and RNAi model. Strong positive correlations ($R = 0.54\sim0.57$) were observed between the screening models (Fig. 3a). Lower correlations ($R = 0.24\sim0.26$) were observed between

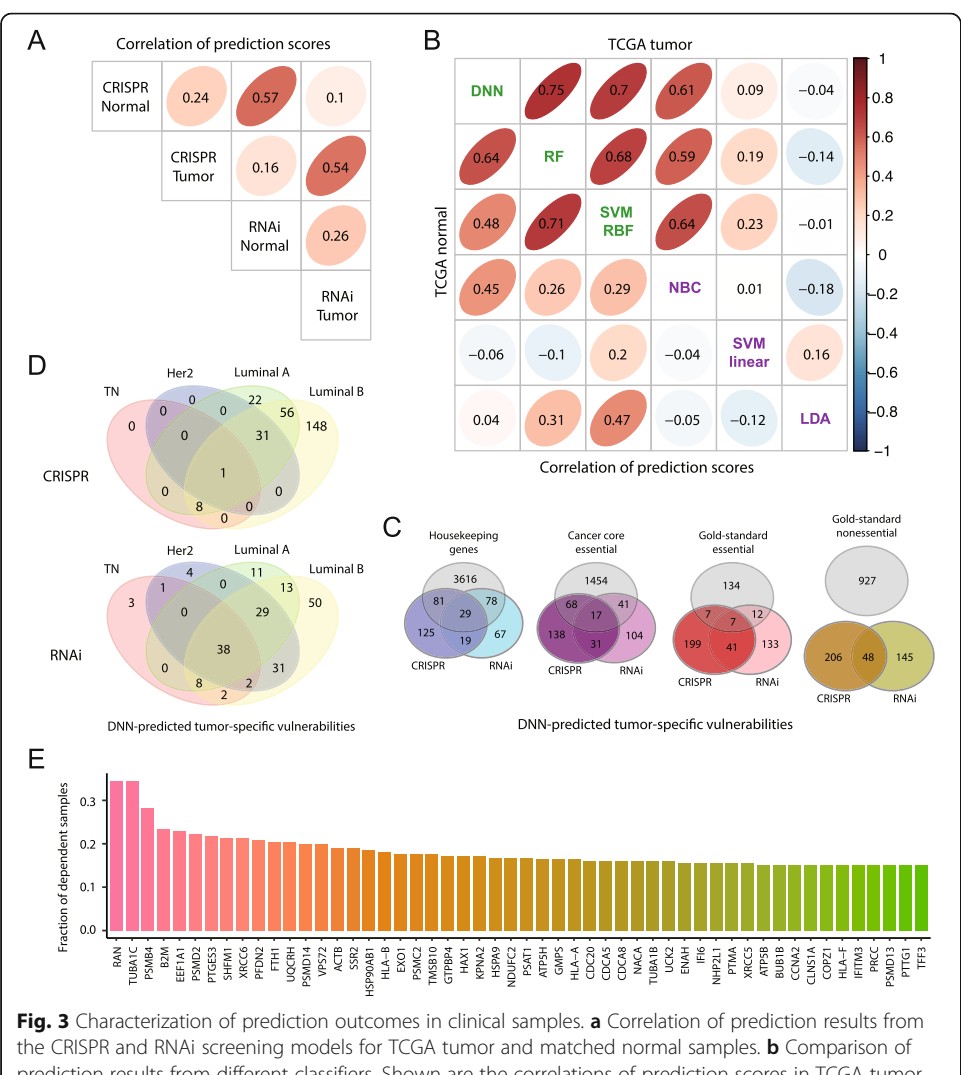

**Fig. 3** Characterization of prediction outcomes in clinical samples. **a** Correlation of prediction results from the CRISPR and RNAi screening models for TCGA tumor and matched normal samples. **b** Comparison of prediction results from different classifiers. Shown are the correlations of prediction scores in TCGA tumor (upper triangle) and matched normal (lower triangle) samples. **c** Intersection of cancer-specific vulnerabilities predicted by DNN with gene lists previously generated from independent studies. **d** Overlap of gene lists presenting tumor-specific vulnerabilities in different breast cancer subclasses. **e** List of genes whose suppression was predicted to confer tumor-specific vulnerability commonly in multiple samples

tumor and normal samples from the same donor (Fig. 3a). We also compared the different classifiers based on the RNAi model. In both tumor and normal samples, the two linear methods, LDA and linear SVM, did not agree with the other classifiers (Fig. 3b). The other linear classifier, NBC, showed positive correlations with the nonlinear classifiers (Fig. 3b). The three nonlinear methods, namely, DNN, radial SVM, and RF, presented highly consistent prediction results (Fig. 3b).

We identified genes that were predicted by DNN as dependencies in tumors but not in normal counterparts. According to a functional enrichment analysis [17], overrepresented functions included proteasome, amino acid biosynthesis, cell cycle, mRNA stabilization, oxidative phosphorylation, and post-translational protein modification (Additional file 3: Table S2). We compared the predicted cancer-specific dependencies with previously defined lists of genes such as the housekeeping genes [18], core fitness genes [1], and gold-standard essential or nonessential genes [19] (Fig. 3c). There were subclass-specific vulnerabilities (Fig. 3d).

We identified genes that were recurrently predicted as cancer-specific dependencies based on the average of the two screening models (Fig. 3e and Additional file 4: Table S3). Up to 35% of the samples were vulnerable to the loss of RAN. RAN encodes the small GTPase Ran, the key regulator of nucleocytoplasmic transport of proteins and ribonucleoproteins. Tumor-specific vulnerability to RAN suppression has been demonstrated in a few cell lines [20]. XRCC6 and XRCC5 encode the Ku heterodimer (Ku70/Ku80), a main component of the nonhomologous end joining (NHEJ) pathway that repairs DNA double-strand breaks (DSBs). Proteasome subunits (PSMB4, PSMD2, PSMC2, PSMD14, PSMD13, and SHFM1) were also included in the list. PTGES3, a co-chaperone functioning with heat shock protein 90 (HSP90), was identified together with other heat shock proteins such as HSP90AB1 and HSPA9. UQCRH, ATP5H, and NDUFC2 are involved in oxidative phosphorylation through respiratory electron transport. We conducted a detailed characterization of dependencies on Ran, Ku70/80, and proteasome subunits in the last section.

We then examined how our predictions for tumor samples differed from those for normal samples. The overall distribution of the dependency scores from different classifiers was compared between tumor and matched normal. In general, tumor samples presented high variability with biases toward high levels of dependencies (Fig. 4a). As discussed in the next section, these results imply that tumor undergoes extensive somatic changes that render the cells vulnerable to the loss of certain molecular activities. Among the nonlinear methods, DNN and radial SVM exhibited this pattern most clearly (Fig. 4a). RF showed this pattern only with the CRISPR model (Fig. 4a).

### Point mutations increase tumor-specific vulnerabilities

Some cancer-specific vulnerabilities should be attributed to somatic DNA lesions that are only present in tumor cells. These can be either loss-of-function (LOF) or gain-of-function (GOF) mutations. When one of the two copies of a gene with an essential function is disrupted by LOF mutations, further suppression of this gene will lead to cell death in only cancer. However, this concept has been examined only for copy number loss in cell lines because matched normal controls are not available [21]. On the

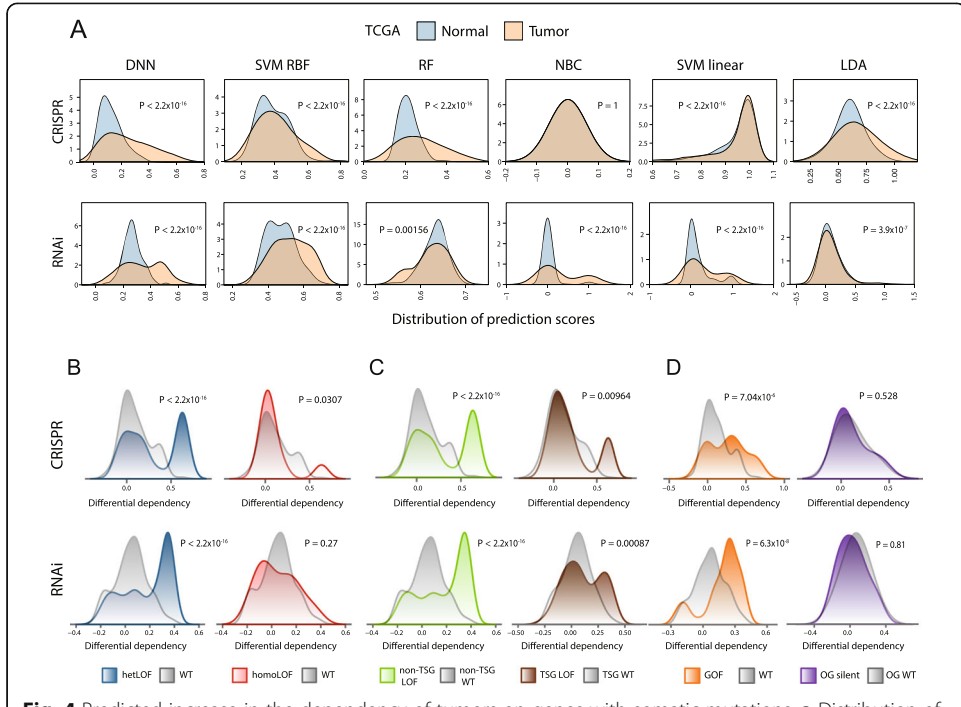

**Fig. 4** Predicted increase in the dependency of tumors on genes with somatic mutations. **a** Distribution of TCGA tumor versus normal prediction scores derived from different classifiers under the CRISPR (upper) and RNAi (lower) screening models. **b** Differential dependency on genes that carry hetLOF (left) or homoLOF (right) mutations in mutant (blue or red) versus wild-type (gray) samples. **c** Differential dependency on non-TSGs (left) or TSGs (right) that carry LOF mutations in mutant (green or brown) versus wild-type (gray) samples. **d** Differential dependency on genes that carry GOF mutations (left) or on known oncogenes (OG) that carry only silent mutations (right) in mutant (orange or violet) versus wild-type (gray) samples. **b–d** P values attached to the density plots are based on the Kolmogorov–Smirnov test. P values over the box plots are from Student's *t* test

other hand, GOF mutations can be used to test oncogene dependence, a phenomenon also referred to as oncogene addiction [22].

We identified heterozygous LOF (hetLOF) and homozygous LOF (homoLOF) mutations. HomoLOF mutations may occur when an inactivation mutation arises in the remaining copy after hemizygous deletion. Also, we detected putative GOF mutations by looking for the cases in which the identical missense substitution was recurrently found across ~ 1000 breast cancer samples. We then segregated the samples according to mutation status. Then, the differential dependency on the mutated gene was obtained by subtracting the normal dependency score from the tumor dependency score.

We wanted to test whether our DNN was able to predict a relative increase in dependency on the mutated genes in tumors carrying the mutations compared to wild-type tumors. First, the differential dependency shifted to positive values specifically in samples with the hetLOF mutations (Fig. 4b, left). This pattern was not observed for samples in which both copies were already inactivated (Fig. 4b, right). Many tumor suppressor genes (TSGs) follow the two-hit hypothesis and contribute to tumorigenesis when both copies are lost or inactivated. Therefore, further suppression of these genes in LOF mutant samples will not cause tumor death. As expected, our DNN method predicted no acquired vulnerabilities for TSGs (Fig. 4c). Acquired dependencies on genes with GOF mutations, but not on oncogenes with silent mutations, were observed

(Fig. 4d). With copy number alterations, there was no distinction between hemizygous versus homozygous deletion, non-TSGs versus TSGs, and oncogenes versus non-oncogenes (Additional file 1: Figure S4). Therefore, point mutations may be a better indicator of acquired vulnerabilities than copy number alterations.

We tested the other classifiers in this respect. The nonlinear classifiers that made similar predictions as DNN, namely, radial SVM and RF, also showed acquired dependencies associated with the LOF or GOF mutations (Additional file 1: Figure S5). The only linear classifier producing results consistent with the nonlinear classifiers, NBC (Fig. 3b), was able to reproduce this pattern as well (Additional file 1: Figure S5). However, DNN showed most clear discrepancies between the mutants and wild-type samples.

## Predictions of context-dependent vulnerability or synthetic lethality

In the previous section, we focused on vulnerabilities pertaining to mutations in the relevant gene itself. Some genes reveal context-dependent vulnerability under certain environmental stress or in particular genetic backgrounds such as cancer subtypes [1, 3]. Synthetic lethality arises when the simultaneous perturbation of two or more genes results in cell death [23]. Here, we examined whether our DNN model could predict context-dependent or synthetic vulnerabilities by focusing on two representative genes identified as common dependencies (Fig. 3e).

We first characterized the tumors that were predicted to be dependent on RAN based on the average of the CRISPR and RNAi model results. First, RAN itself and some RAN-related genes such as RANGAP1, RANBP1, RCC1, and TPX2 were overexpressed in these samples (Fig. 5a). Second, Her2-positive tumors, which grow faster than other subtypes, were significantly associated with RAN dependency (Fig. 5a). Third, functional enrichment of the overexpressed genes in the RAN-dependent samples was found for nuclear chromatin, ribosome, and spliceosome (Fig. 5a and Additional file 5: Table S4). In rapidly dividing tumors, proteins involved in chromatin formation need to be actively synthesized and transported into the nucleus. The assembly of ribosomes and spliceosomes is reliant on nucleocytoplasmic transport as the proteins are imported into the nucleus and then exported back to the cytosol after coupling with the RNAs. Ran is activated by growth signaling to fulfill this demand [24]. Therefore, rapidly growing tumors should be vulnerable to RAN silencing.

XRCC6 and XRCC5 encode Ku70 and Ku80 proteins, respectively. The Ku heterodimer recruits DNA-PKcs (encoded by PRKDC) to DSBs for DNA repair by the classical NHEJ pathway [25]. Upregulation of XRCC6 and other core NHEJ factors (e.g., PRKDC, LIG4, and XRCC4) was observed in the Ku-dependent tumors (Fig. 5b). Particularly significant upregulation was observed for ASF1A (Fig. 5b). This gene interacts with MDC1 to promote NHEJ repair [26]. MDC1 was also overexpressed (Fig. 5b). Two other mechanisms to repair DSBs are homologous recombination (HR) and alternative NHEJ. There is a synthetic lethal relationship between HR and classical NHEJ. HR-defective cells are sensitive to DNA-PKcs inhibition [27, 28]. MSH3 was identified as a novel HR pathway gene that significantly increases the sensitivity [28]. We observed a significant copy number loss of MSH3 and major HR genes (BRCA1, RAD50, and RAD51) in the Ku-dependent samples (Fig. 5b). Activation of alternative NHEJ was

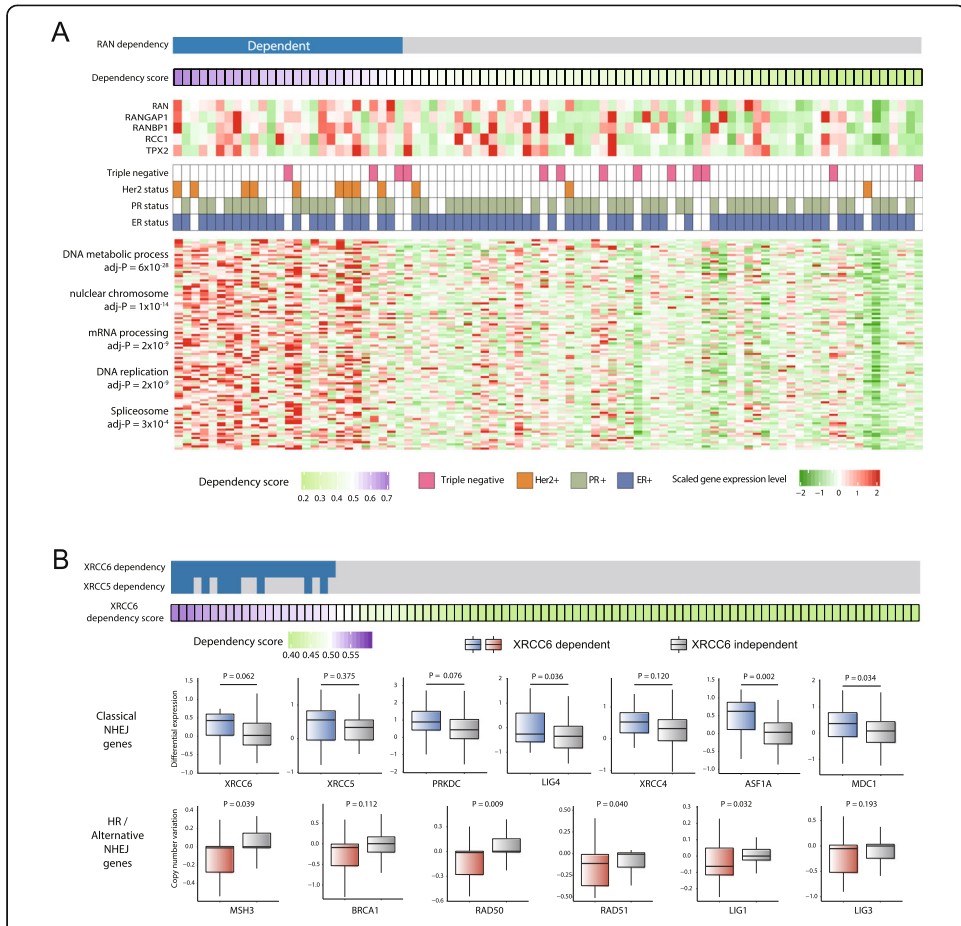

**Fig. 5** Characterization of context-dependent or synthetic vulnerabilities. **a** Analysis of RAN-dependent samples. Dependency of each sample (column) as estimated by the DNN model is indicated at the top. Expression levels of RAN-related genes are shown below. The triple-negative subtype was determined by each sample's status for estrogen receptor (ER), progesterone receptor (PR), and human epidermal growth receptor-2 (Her2). The heatmap below presents the transcription profile of the most significant 100 genes that were overexpressed in the RAN-dependent samples ($P < 0.005$ by the two-sample $t$ test). Enriched functional terms are indicated on the left. **b** Analysis of Ku70/80-dependent samples. DNN-estimated dependency on XRCC6 or XRCC5 is marked in blue. Differential expression of the classical NHEJ genes in tumor relative to normal samples is shown. Also shown are the gene-level copy number scores of MSH3 and some genes implicated in HR (BRCA1, RAD50, and RAD51) or alternative NHEJ (LIG1 and LIG3). Negative values indicate copy number loss. $P$ values are from Fisher's exact test for the overrepresentation of copy loss samples in the dependent group

not observed in these tumors; whereas PARP1, XRCC1, MRE11, and RBBP8 (encoding CtIP) did not show upregulation (Additional file 1: Figure S6), some genes (LIG1 and LIG3) exhibited copy number loss (Fig. 5b).

### Experimental validation of predicted dependencies

To test the applicability of our method in the clinical setting, we performed DNN prediction on the transcriptomes of our 24 patient-derived xenografts of breast cancer [29]. For RAN, 23 of the 24 tumors were predicted to be dependent. For experimental validation, we selected two samples whose predicted dependencies were positive on RAN and named them PD(+)RAN-1 and PD(+)RAN-2. For each of XRCC6 and PSMB4, we chose two

samples with positive predicted dependencies and two with negative predicted dependencies: PD(+/−)XRCC6/PSMB4-1/2. We derived patient-derived cells for these samples; performed CRISPR-Cas9 [30] targeting of RAN, XRCC6, or PSMB4; and confirmed the stable expression of Cas9 and the reduced expression of the target genes.

RAN suppression in the PD(+)RAN cells resulted in an increase in the proportion of Annexin V-positive cells, indicating the induction of apoptosis, and a decrease in the fraction of proliferating cells according to the BrdU assay (Fig. 6a and Additional file 1: Figure S7). For XRCC6 and PSMB4 silencing, the induction of apoptosis and the decrease of cell proliferation were remarkably more pronounced in the PD(+) cells than in the PD(−) cells (Fig. 6b, c and Additional file 1: Figures S8~S9). These results demonstrate that our method to predict sample-specific vulnerabilities can lead to the discovery of precision therapeutic targets.

Next, we explored the mechanisms underlying cell death induced by the inactivation of these genes. In the PD(+)RAN cells, we tested the nuclear translocation of epidermal growth factor receptor (EGFR). The ligand EGF stimulates the translocation of phosphorylated EGFR (pEGFR) to the nucleus [31]. The EGFR nuclear transport was reported to be RAN-dependent [32]. When RAN was inactivated by CRISPR/Cas9, the nuclear fraction of pEGFR vanished whereas cytoplasmic pEGFR was not affected (Fig. 7a, b and Additional file 1: Figure S10). This resulted in the downregulation of two important transcriptional targets of pEGFR, that is, Cox2 and Cyclin D1 (Fig. 7a). For PSMB4, we tested the degradation of proapoptotic proteins. PSMB4 knockdown increased the protein levels of Bad, Bim, and Cytochrom C in the PD(+)PSMB4 cells (Additional file 1: Figure S11).

Regarding Ku70/80 dependency, we first tested whether the PD(+)XRCC6 cells rely on the NHEJ pathway. When XRCC6 was inactivated, the PD(+) cells maintained the activity of DNA-PKcs whereas the PD(−) cells decreased the expression of this protein (Additional file 1: Figure S12). Indeed, the PD(+) samples were more sensitive to the DNA-PK inhibitor, NU-7026, as indicated by 40–60% lower IC50 values (Additional file 1: Figure S13A~S13B). This reliance on the NHEJ pathway may reflect HR deficiency. It is known that HR-defective cells are sensitive to DNA-PKcs inhibition [27, 28]. To test this, we challenged the PD(+/−)XRCC6 cells with γ-irradiation to generate DSBs. XRCC6 inactivation led to an increase in the number of Rad51 foci in both the PD(+) and PD(−) cells (Fig. 7c). However, the degree of the increase of Rad51 foci was 3~4-fold lower in the PD(+) cells than in the PD(−) cells (Fig. 7d). We also performed the DR-GFP/I-*Sce*I HR assay [33] in which the fraction of GFP-positive cells indicates HR activity for DSB repair. While XRCC6 suppression led to an increase in the fraction of the GFP-positive cells (Fig. 7e), the magnitude of increase was 3~6-fold lower in the PD(+) cells than in the PD(−) cells (Fig. 7f). The similar results were obtained from the DR-GFP/I-*Sce*I assay in response to the treatment of NU-7026 (Additional file 1: Figure S13C). In summary, the PD(+)XRCC6 cells displayed HR deficiencies under high DSB load, consistent with our observation with the TCGA data (Fig. 5b). These results illustrate how Ku70/80 silencing can render particular tumors vulnerable when the sensitivity to Ku70/80 suppression can be predicted by our computational method.

## Discussion

There was an attempt to develop a predictive model on the basis of cell line screening data [4]. This model was reliant on common molecular markers of cell lines with

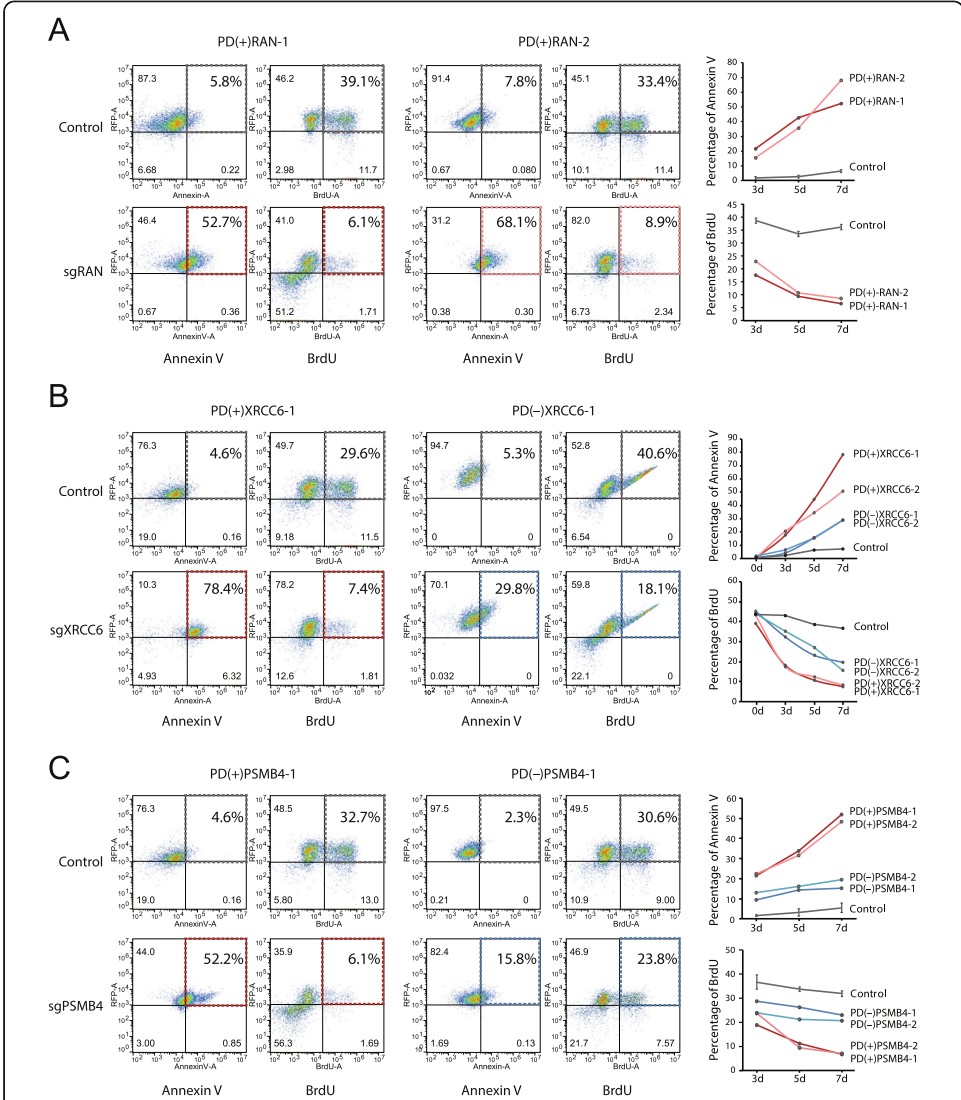

**Fig. 6** Experimental validation of predicted vulnerabilities by apoptosis and cell proliferation measurements. **a** Scatter plots showing the distribution of Annexin V and BrdU staining, which marks apoptosis and cell proliferation, respectively, for two patient-derived cells with positive predicted dependency on RAN, namely, PD(+)RAN-1 and PD(+)RAN-2, before and after RAN inactivation by CRISPR-Cas9. Shown here are day 7 results. Plots for days 3 and 5 are provided in Additional file 1: Figure S7. The graphs on the right indicate the percentage of Annexin V- or BrdU-positive cells according to days. **b**, **c** Scatter plots showing the distribution of Annexin V and BrdU staining for patient-derived cells with positive or negative predicted dependency on **b** XRCC6 and **c** PSMB4 before and after CRISPR-Cas9 knockout. Shown here are day 7 results for PD(+)XRCC6-1, PD(−)XRCC6-1, PD(+)PSMB4-1, and PD(−)PSMB4-1. Plots for days 3 and 5 after inactivation in these cells, and data for days 3, 5, and 7 after inactivation in PD(+)XRCC6-2, PD(−)XRCC6-2, PD(+)PSMB4-2, and PD(−)PSMB4-2 are provided in Additional file 1: Figures S8~S9. The graphs on the right indicate the percentage of Annexin V- or BrdU-positive cells according to days after inactivation

shared dependencies. In this case, the number of available cell lines for each tumor type, and thus the number of dependent cases to be learned, is limited. Moreover, this approach is applicable only to context-dependent variability or synthetic lethality, and predictions cannot be made for sample-specific dependencies.

Instead of using basal expression patterns as predictors, we performed simulations to obtain transcriptome patterns resulting from gene suppression. Because these

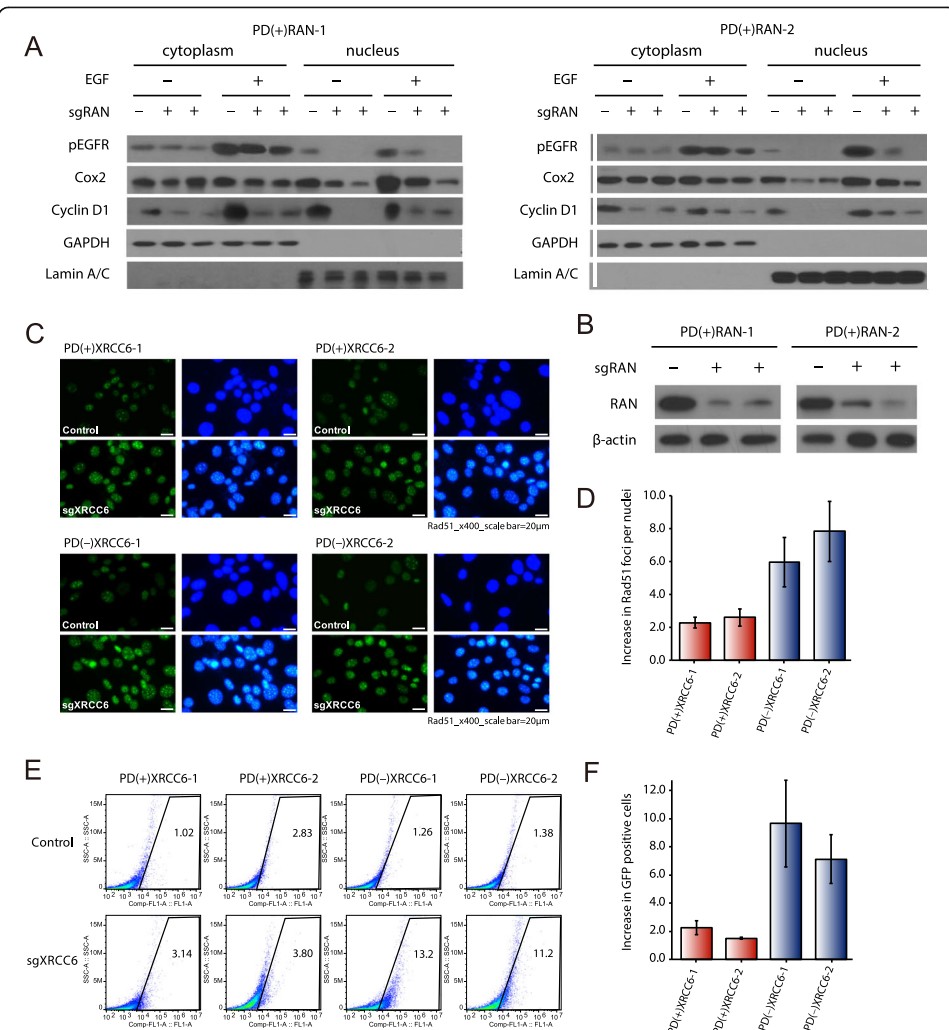

**Fig. 7** Functional studies of RAN and XRCC6 inactivation. **a** Western blot analysis of nuclear/cytoplasmic fraction in PD(+)RAN-1 and PD(+)RAN-2 before and after RAN silencing. The nuclear translocation of pEGFR and its transcriptional targets (Cox2 and Cyclin D1) was examined. GAPDH and Lamin A/C were used as a cytoplasmic and nuclear specific loading control, respectively. **b** Confirmation of RAN suppression by two sgRNAs (sgRAN-1 and sgRAN-2) in two cells (PD(+)RAN-1 and PD(+)RAN-2). **a, b** Images of the full uncropped scans are provided in Additional file 1: Figure S10. **c** Immuno-fluorescence images for Rad51 (green) in irradiated PD(+)XRCC6 (upper) and PD(−)XRCC6 (lower) cells before and after XRCC6 inactivation (control vs sgXRCC6). DAPI staining results (blue) are shown side by side. **d** A graph showing the ratio (sgXRCC6/control) of the number of Rad51 foci per cells. The mean and standard error were obtained from 5 fluorescence images. **e** Scatter plots showing the rate of HR repair based on the DR-GFP/I-*Sce*I assay in irradiated PD(+)XRCC6 and PD(−)XRCC6 cells. GFP-positive cells in the marked zones indicate a population that underwent HR-mediated DSB repair of GFP reporter plasmids. **f** A graph showing the ratio (sgXRCC6/control) of the GFP-positive cells. The mean and standard error were obtained from 5 experiments after excluding the maximum and minimum

simulations could be repeated for individual genes, as many perturbed profiles as genes could be obtained from one basal profile, which cannot be accomplished experimentally. Therefore, a sufficient number of cases could be used to train predictive models. Also, it is difficult to obtain experimental profiles associated with cell death because the expression patterns of only living cells can be captured.

The three genes we tested experimentally are presented as common essential genes in the depmap portal browser (https://depmap.org/portal). In contrast, our model

predicts that they are dependencies in 20~30% of tumors (Fig. 3e). As for XRCC6 and PSMB4, this seems to result from the different scoring scheme used in the browser. The scoring metric used in our model indicates similar fractions of dependent cell lines at the threshold we used, thereby emphasizing consistencies between our prediction and cell line screening results. As for RAN, we characterized the gene as a context-dependent dependency in association with cell growth (Fig. 5a). This may explain why RAN is detected as a common dependency in cell line screening in which rapidly growing cells dominate the culture population. This highlights the advantage of our model over screening results.

A classic example of synthetic lethality is the interaction between poly (ADP-ribose) polymerase 1 (PARP1) and BRCA1/2 [34–36]. Later, the sensitivity of BRCA1/2-mutated tumors to PARP inhibitors was broadly linked to HR repair defects [23]. Further studies established synthetic lethality between the HR pathway and the NHEJ gene DNA-PKcs [27, 28, 37, 38]. Our results suggest that HR-defective tumors are also sensitive to the loss of Ku70/80, the interacting partner of DNA-PKcs. Further investigation is required as to whether inhibiting Ku70/80 instead of DNA-PKcs can be more effective at the RNA level. Importantly, sensitivity to Ku70/80 inhibition can be predicted based on transcriptome data by our deep learning model.

This approach can be extended to many different cancer types, facilitating the discovery of precision therapeutic targets. Once a prediction model has been established, exome and transcriptome data for tumor and matched normal samples will be sufficient to identify patient-specific vulnerabilities that are associated with somatic mutations or common vulnerabilities that arise in a context-dependent or synthetic lethal manner.

## Materials and methods

### Cell line dependency screening data

We downloaded two independent CRISPR-Cas9 dropout screening results for 28 and 25 breast cancer cell lines [9, 10] from https://depmap.org/portal/download and https://score.depmap.sanger.ac.uk/downloads. Genetic dependencies were measured based on the CERES [9] and BAGEL [15] scores, respectively. We selected different cutoff values (CERES = − 0.6, − 1.0, or − 1.5 and BAGEL = 0, 2, or 4) to result in the similar number of dependencies per cell line between the two datasets. In addition, we obtained shRNA dropout screening results for 77 breast cancer cell lines [3] from http://neellab.github.io/bfg. Normalized GARP score, or zGARP, was used as a metric of dependency [3, 16]. Here, a zGARP of − 2, − 3, or − 4 was used as the threshold to define experimentally determined dependency. Gene expression profiles of the cell lines were also obtained from the same websites.

### Regulatory network construction

For regulatory network construction, we downloaded TCGA RNA-seq data for 1215 breast cancer and 423 liver cancer samples from the UCSC Cancer browser (https://xenabrowser.net). We employed our previous Bayesian network in breast cancer [11] (Additional file 6: Table S5) and the same type of network in liver cancer [14] for a negative control. We constructed another type of regulatory network on the basis of ARACNe (Algorithm for the Reconstruction of Accurate Cellular Networks) [12]

(Additional file 7: Table S6). We applied the ARACNe software available at http://califano.c2b2.columbia.edu/aracne [39]. From a gene expression profile, ARACNe identifies direct transcriptional interactions on the basis of mutual information (MI). We determined the significance of MI at $P = 0.01$. In order to simulate the effect of gene suppression, we had to determine the directionality (i.e., which gene regulates which gene) and regulation type (i.e., whether the interaction is activation or inhibition) of the interactions between the suppression target gene and its interaction partners. For this purpose, we utilized conditional probability. For example, when $P(X = \text{on} | Y = \text{on})$ represents the probability that gene X is expressed given that gene Y is expressed, the directionality can be determined by comparing $P(X = \text{on} | Y = \text{on})$ and $P(Y = \text{on} | X = \text{on})$. If $P(X = \text{on} | Y = \text{on}) > P(Y = \text{on} | X = \text{on})$, gene Y is considered to be a regulator of gene X. To determine the regulation type, we used $P(X = \text{up/down} | Y = \text{up/down})$ to represent the probability of gene X being up- or downregulated given the up- or downregulation of gene Y. More details are provided in the next section.

### In silico CRISPR/RNAi methodologies

The first step of our in silico CRISPR/RNAi was to identify genes whose expression level would be affected by the suppression of the target gene. From the Bayesian network, all descendant genes were identified by traveling via outgoing links from the target gene. We determined only whether a given gene is a descendant of the target gene while ignoring the structure of the intervening nodes between the two genes. Then, the expression level of each descendant was adjusted depending on the sign and strength of its expression correlation with the target gene. Here, we denote the target gene as gene Y and the $j$th descendant of gene Y as gene $X_j$. When $x_j$ and $y$ indicate the basal expression level of $X_j$ and Y in the given sample, respectively, the perturbed expression level of the $j$th gene, $x'_j$, can be obtained as

$$x'_j = x_j - r_j \frac{y - y'}{y} x_j,$$

where $r_j$ is the correlation coefficient of the expression levels of gene $X_j$ and Y across reference samples, and $y'$ is the perturbed expression level of the target gene, Y. For reference samples, we used the same transcriptome data that were used for network construction. We set $y' = 0$ for CRISPR simulation and $y' = 0.2y$ for RNAi considering the known average experimental efficacy of shRNA [40]. Instead of repeating this calculation gene by gene, we utilized matrix multiplication. The perturbed expression level of the $j$th gene by the $i$th target gene was obtained as the $(i,j)$th entry of the perturbed expression matrix $P$ as follows.

$$P_{i,j} = -0.8(R{\cdot}B)_{i,j} + B_{j,j}$$

$$\text{where } R = \begin{bmatrix} 1 & \cdots & r_n \\ \vdots & \ddots & \vdots \\ 0 & \cdots & 1 \end{bmatrix} \text{ and } B = \begin{bmatrix} x_1 & \cdots & 0 \\ \vdots & \ddots & \vdots \\ 0 & \cdots & x_n \end{bmatrix}$$

$R$ is an adjacency matrix for expression correlations and $B$ is a basal expression matrix filled with zero except at the diagonal.

For the ARACNe network, we utilized the conditional probability instead of the correlation coefficient. First, all the first neighbors of the target gene were identified. We denote the target gene as gene Y and the $j$th neighbor of gene Y as gene $X_j$. Among them, the regulatory targets, not the regulators, of gene Y were determined based on the following set of conditional probabilities.

$$P(X_j = \text{activator}) = \frac{P(Y = \text{up} \cap X_j = \text{up}) + P(Y = \text{down} \cap X_j = \text{down})}{P(X_j = \text{up}) + P(X_j = \text{down})}$$

$$P(Y = \text{activator}) = \frac{P(X_j = \text{up} \cap Y = \text{up}) + P(X_j = \text{down} \cap Y = \text{down})}{P(Y = \text{up}) + P(Y = \text{down})}$$

$$P(X_j = \text{inhibitor}) = \frac{P(Y = \text{down} \cap X_j = \text{up}) + P(Y = \text{up} \cap X_j = \text{down})}{P(X_j = \text{up}) + P(X_j = \text{down})}$$

$$P(Y = \text{inhibitor}) = \frac{P(X_j = \text{down} \cap Y = \text{up}) + P(X_j = \text{up} \cap Y = \text{down})}{P(Y = \text{up}) + P(Y = \text{down})}$$

Up- and downregulation was defined using the reference transcriptome samples that were used for network construction. For each gene, the mean expression level, $\mu$, and its standard deviation, $\sigma$, were obtained across the reference samples. We set $X_j = \text{up}$ and $Y = \text{up}$ when their expression level in the given reference sample was larger than $\mu + \sigma$. On the contrary, we set $X_j = \text{down}$ and $Y = \text{down}$ when their expression level in the given reference sample was lower than $\mu - \sigma$. $X_j$ was considered to be a regulatory target of Y when $P(X_j = \text{activator}) + P(X_j = \text{inhibitor}) < P(Y = \text{activator}) + P(Y = \text{inhibitor})$. Once $X_j$ has been determined as a target of Y, the regulation type of the link from Y to $X_j$ was determined by comparing $P(Y = \text{activator})$ and $P(Y = \text{inhibitor})$. Finally, when $x_j$ and $y$ indicate the basal expression level of $X_j$ and Y, respectively, in a given sample subjected to in silico CRISPR/RNAi, the perturbed expression level of the $j$th gene, $x'_j$, was obtained as

$$x'_j = \begin{cases} x_j - P(Y = \text{activator})\dfrac{y - y'}{y}x_j, \text{ if } P(Y = \text{activator}) > P(Y = \text{inhibitor}) \\ x_j + P(Y = \text{inhibitor})\dfrac{y - y'}{y}x_j, \text{ if } P(Y = \text{activator}) < P(Y = \text{inhibitor}) \end{cases}$$

For CRISPR/RNAi target genes, we considered all genes in the network, except those at the terminal nodes, when the expression data was available in the basal transcriptome that is subjected to in silico CRISPR/RNAi. We repeated the above procedures for all available genes. In a test for the influence of the in silico CRISPR/RNAi processes on prediction performance, we chose genes that have > 100 descendants or regulatory targets in the network and used their perturbed expression patterns as training input. We performed network randomization by shuffling the nodes while maintaining the network structure. We also generated inverted networks by reversing the directions of all links to test the validity of inferred regulator-target relationships.

In breast cancer, the Bayesian regulatory network consisted of 13,047 genes, and the ARACNe network was composed of 8463 genes. Among them, 7439 and 7086 genes in the Bayesian network and ARACNe network, respectively, turned out to regulate at least one gene. Therefore, we were able to perform in silico CRISPR/RNAi for these genes only.

## Deep neural network modeling

We developed a deep neural network (DNN) model by using the Theano library [41]. DNN architecture expresses complex nonlinear relationships embedded in input features through multiple layers of artificial neural networks. However, a large number of manual attempts needs to be tested because model performance is sensitive to many calibration parameters or hyperparameters. Here, we used three types of calibration parameters: (i) parameters for model architecture, such as the number of hidden layers and number of hidden nodes; (ii) parameters for model algorithm, such as the learning rate, momentum, batch size, activation function, and initial weight distribution; and (iii) parameters to avoid overfitting, such as regularization parameters (L1 and L2) and dropout rate. The calibration parameters we used in this work are summarized in Additional file 2: Table S1.

Pre-training processes can lead to optimal initial points of weight and bias, thereby reducing the chance to fall into local minima [42]. We constructed a stacked denoising autoencoder (SdA) with the same structure of the DNN model except that the output layer has the same number of nodes as the input layer. Through a process known as denoising, we generated a stochastically corrupted version of the log-normalized input vector x consisting of the perturbed expression levels of $n$ genes, $x \in [0, 1]^n$. The SdA mapped the corrupted x to a hidden layer y, $y \in [0, 1]^m$, using the activation function $f$. This encoding process can be expressed as

$$y = f(Wx + b),$$

where W is the weight matrix and b is the bias. The following decoding step generated the reconstruction vector z given as

$$z = f(W^T y + b')$$

while minimizing reconstruction error represented as Cost. Cost was defined as follows according to the activation function of the DNN model:

1. If the ReLU function was used,

$$Cost = \frac{1}{B}\sum_{k=1}^{B}(x_k - z_k)^2.$$

2. If the sigmoid function was used,

$$Cost = -\frac{1}{B}\sum_{k=1}^{B}[x_k \log z_k + (1 - x_k) \log(1 - z_k)].$$

Here, $B$ indicates a batch size. Some entries of the input vector x were masked according to the dropout rate. Parameter $\theta$ (weight and bias) was updated through the stochastic gradient descent at each training epoch step as

$$\theta_{t+1} = \theta_t - \alpha \nabla_{\theta_t},$$

where $t$ indicates training epochs.

To further prevent overfitting, we applied early stopping and feature selection.

1. Early stopping is a form of regularization used to avoid overfitting by stopping training at points where performance does not improve. One fifth of the training

data were used for performance evaluation. Three fourths of the remaining were used for model learning, and the remaining 1/4 were used for stop control. All models were trained only to the point when performance reached a plateau. To determine the stopping point while avoiding overfitting, the training set was further divided into two independent parts, one for model training and the other for performance tracing. We defined a variable called *patience* and initially set it to 200. We then trained the model while keeping the number of training epochs below *patience*. Whenever performance improvement was observed, we updated *patience* to max(initial *patience*, current epoch × 2) and saved the current model as the best. The threshold of improvement was set to 0.99.

2.  To prevent overfitting due to an excess of learning features, we reduced the number of genes whose perturbed expression levels were used as input. Specifically, we selected 1000 genes whose expression values varied most across different cases (perturbed transcriptomes) based on standard deviation.

After pre-training, we fine-tuned the model to minimize the loss function defined as

$$\text{Loss} = \text{NLL} + \lambda_1 ||w||_1 + \lambda_2 ||w||_2,$$

where NLL denotes the mean of negative log likelihood, and $\lambda_1 ||w||_1 + \lambda_2 ||w||_2$ represents the regularization term of the elastic net that is used to control overfitting. $||\cdot||_p$ indicates the $L_p$ norm and was given as

$$||w||_p = \left( \sum_{j=0}^{|w|} |w_j|^p \right)^{\frac{1}{p}},$$

where $\lambda_p$ is a hyperparameter that controls the relative contribution of each regularization term. As implied in the above formula, the elastic net is a combination of lasso ($L_1$) and ridge ($L_2$) regularization on the weights of the model. The elastic net is known to outperform $L_1$ or $L_2$ regularization alone and to be especially useful when the number of predictors is large [43]. The $NLL(\theta)$ of the loss function was given as

$$NLL(\theta) = -\frac{1}{B} \sum_{i=1}^{B} \left( Y^i \log f(\theta)^i + \left(1 - Y^i\right) \log \left(1 - f(\theta)^i\right) \right),$$

where $f(\theta)^i$ is an output for the $i$th in silico CRISPR/RNAi target gene in a mini-batch of size $B$. Each target, $Y$, can be either 1 or 0 where 1 indicates that there is dependency on the gene in the given sample. The loss function was composed of parameters ($\theta$) including $w_m^k$, $b_k$, $w_k$, and $b'$, which were updated by the standard backpropagation algorithm with momentum. The momentum method is a technique to accelerate the gradient descent process by accumulating a velocity (v) in the direction of stable and persistent reduction of the loss function [44]. For the given loss function, the momentum was given as

$$\theta_{t+1} = \theta_t + v_{t+1},$$

$$v_{t+1} = \mu v_t - \varepsilon \nabla \left( \text{Loss}\left(\theta^t\right) \right),$$

where $\varepsilon$ is the learning rate, $\mu$ is the momentum coefficient, and $\nabla(\text{Loss}(\theta^t))$ is the gradient at $\theta^t$. $v_0$ was set to 0.

## Model construction

While evaluating prediction performance on a total of 93,312 combinations of hyper-parameter values (Additional file 2: Table S1), we observed that performance does not improve considerably after 1000 trials in general. Therefore, we used 1000 random combinations of the parameter values for model selection. The learning processes were based on five-fold cross-validation consisting of five different partitions into training and validation data. For each partition of training and validation data, the best hyper-parameter combination was selected based on the AUC on the validation data set. A test AUC was then assigned to the selected model based on an independent test set that was not used for learning. The average test AUC for the five best models was obtained to represent the overall performance. In our tests of the shuffled, inverted, and liver cancer networks, we selected the five best hyperparameter combinations for each split into a training set and a validation set, resulting in five average test AUC values. The mean of these five values was reported. We used the class probability as the "prediction score" for the given input. A prediction score close to 1 indicates a higher association of the given transcriptome with cell death. The input transcriptome with a prediction score greater than 0.5 was classified as a dependency profile. Tumor-specific dependencies were identified as having the tumor prediction score greater than 0.5 and the normal prediction score lower than 0.5.

## Other classifiers

We also implemented random forest [45], support vector machine [46], naïve Bayes classifier [46], and linear discriminant analysis [47]. We tested the linear, polynomial, sigmoid, and radial basis kernels for the support vector machine. By using the C-classification type, we used the support vector machine as a classification machine. Random forest was trained with 1000 decision trees by using the R package "random-Forest." We used the R package "e1071" to implement support vector machine and naïve Bayes classifier. The R package "MASS" was used to implement linear discriminant analysis. A prior probability of 0.5 was given to the two classes (dependency versus independency).

## Analysis of TCGA breast cancer data

We downloaded the expression profiles of 1095 tumor samples along with those of 113 matched normal samples from the TCGA database [13]. Somatic variant calls from whole-exome sequencing for 988 cases were also obtained. Of these, the matched normal expression profiles were available for 110 samples. For copy number analysis, we obtained the gene-level GISTIC2.0 scores [48], which were available for 111 samples. Copy number variations for genomic segments were mapped to protein-coding genes by the TCGA FIREHOSE pipeline. The gene-level GISTIC2.0 scores were classified into homozygous deletion, heterozygous deletion, normal copy, low-level amplification, and high-level amplification. We used homozygous/heterozygous deletions and low/high-level amplifications unless otherwise stated (e.g., hemizygous deletion). Immunohisto-chemistry data for estrogen receptor, progesterone receptor, and human epidermal growth receptor-2 were available for 88 samples. The triple-negative subtype was

determined by each sample's status for estrogen receptor, progesterone receptor, and human epidermal growth receptor-2.

The prediction of the dependencies of the TCGA samples was performed using in silico CRISPR/RNAi based on the Bayesian network. We used the 113 sample for which matched normal transcriptomes were available. We compared the prediction scores from the DNN model, random forest, support vector machine with radial basis function or linear kernels, naïve Bayes classifier, and linear discriminant analysis. Tumor-specific dependencies were identified when the DNN prediction score was greater than 0.5 for the tumor sample and lower than 0.5 for the matched normal sample.

### Analysis of dependency-associated somatic alterations

We collected known oncogenes and tumor suppressor genes (TSGs) from the Cancer Gene Census database [49] and based on the 20/20 rule [50]. In addition to the known cancer genes, we benchmarked the 20/20 rule to discover putative oncogenes and TSGs in breast cancer. For this purpose, we used somatic variant calls from whole-exome sequencing for 988 breast cancer samples. We first catalogued a set of inactivation mutations, which were defined as nonsense mutations or frameshift insertions and deletions. Genes carrying the inactivation mutations in five or more breast cancer samples were regarded as putative TSGs. The known TSGs and putative TSGs were combined and referred to as TSGs in the "Results" section. Putative oncogenes were identified based on missense variants. Specifically, genes carrying the same substitution at the identical position in two or more samples were regarded as potential oncogenes in breast cancer.

For the analysis of dependency-associated mutations, we used somatic variant calls for the 110 tumors that came with tumor and matched normal transcriptomes. For loss-of-function (LOF) mutations, we used nonsense mutations and frameshift insertions or deletions. To detect heterozygous LOF (hetLOF) mutations, we identified heterozygous calls with the variant allele frequency < 40%. Homozygous calls or heterozygous calls with the variant allele frequency > 50% were regarded as homozygous LOF (homoLOF) mutations. Gain-of-function (GOF) mutations were defined as the recurrent missense variants that we detected to identify the putative oncogenes as described above. The known oncogenes that carried silent mutations only were used as negative controls. For this purpose, we excluded the known oncogenes that have any nonsynonymous mutations in them. The effect of copy number alterations was examined after filtering genes that carry any LOF or GOF mutations in them. The effect of copy number amplifications and silent mutations was examined for the known oncogenes while excluding the putative oncogenes.

### Construction of CRISPR/Cas9 vector and Cas9-expressing cells

Lentiviral guide RNA plasmids were constructed with the pLKO.sgRNA.EFS.tRFP (Addgene #57823) backbone. Two small guide RNAs (sgRNAs) were designed for each of RAN, XRCC6, and PSMB4 and were inserted into the vector using BsmB I. The two sgRNAs were used simultaneously except for the functional studies of RAN and PSMB4 knockout. The sequences of the sgRNAs used are in Additional file 8: Table S7.

To produce Cas9-expressing lentiviruses, HEK 293T cells in 100-mm plates with 70% confluency were co-transfected with packaging plasmids pMD2.G (10 μg) and psPAX2 (10 μg), lentiCas9-Blast (Addgene #82372), and pLKO.sgRNA.EFS.tRFP (Addgene #57823), using the calcium-phosphate transfection method. After 12 h, medium was changed to DMEM medium with 10% FBS and 1% PS. Every 12 h thereafter, the culture supernatant containing the viral particles was harvested and subjected to centrifugation at 1700 rpm at 4 °C for 10 min so as to remove any remaining HEK 293T cells.

To generate primary breast cancer cells that stably express Cas9, the culture supernatant containing the viruses was supplemented with polybrene (1 μg/ml) and administered to cancer cells for 12 h. This process was repeated three times. After 24-h incubation in fresh RPMI media, the stable Cas9-expressing cells were obtained by selection with 4~8 μg/ml of Blasticidine S hydrochloride (Sigma).

### Knockdown in patient-derived primary cells

Patient-derived primary cells were cultured in RPMI media containing 5% FBS, 1% penicillin/streptomycin, hEGF, hydrocortisone, and transferrin. Viruses with control- or target gene-sgRNA were prepared and applied to infect the primary cancer cells as described above for the construction of Cas9-expressing cells. To confirm knockdown of target genes, cells were harvested 24 h after the last viral treatment and analyzed by the real-time PCR. Briefly, total RNA extraction was performed using TRizol (Invitrogen). Next, 1 μg of total RNA was subjected to cDNA synthesis (PrimeScript RT reagent kit, Cat No. PR037A, Takara). The primer sequence for each target is as follows: Cas9 (forward 5′-TTGAAAGGAGTGGGAATTGG-3′, reverse 5-′CACGGCGACTACTGCACTTA-3′), XRCC6 (forward 5′-CTTTGAGGAATCCAGCAAGC-3′, reverse 5′-ATTGGAGGAG GCTTGAGAGC-3′), RAN (forward 5′-TGTTCCACACCAACAGAGGA-3′, reverse 5′-TGTTGCCACACAACACAATG-3′), and PSMB4 (forward 5′-GCCAGATGGTGATT GATGAG-3′, reverse 5′-CATCGGAGGCTATGCTGATG-3′). Relative expression value was normalized by hRPL13a.

### BrdU incorporation and Annexin V apoptosis assay

To examine cell cycle progression, the patient-derived cells transfected with the CRISPR/Cas9 lentiviral vector were seeded onto a 6 well-plate. After 24 h, the cells were infected with lentiviruses expressing either control (pLKO5-tRFP; Addgene 57823) or sgRNAs (pLKO5-sgRNAs-tRFP). At each time point (3, 5, or 7 days after lentiviral transfection), 10 μM of BrdU was added and incubated for 6–8 h. In vitro BrdU kit (BD Pharmingen, San Diego, CA) was used as described by the manufacturer. Briefly, the cells were fixed with ethanol-containing fixation buffer and washed with the Perm/Wash buffer. For BrdU staining, cells were labeled by the FITC-conjugated anti-BrdU antibody for 20 min at room temperature. After incubation, cells were washed and resuspended with the FACS staining buffer. Finally, FITC (BrdU) and RFP (sgRNA)-positive cells were measured using AccuriFlow Cytometry (BD Biosciences). The percentage of the FITC and RFP-positive cells was calculated using the CFlow software. For apoptosis assay, the cells were seeded onto a 6-well plate as mentioned above. Apoptotic cells were measured by the FITC-Annexin V Apoptosis Detection Kit (BD Pharmingen) at the same time point following the instructions of the manufacturer.

Briefly, cells were incubated with FITC-Annexin V in the FACS staining buffer for 15 min. The stained cells were immediately subjected to the CFlow software analyses using AccuriFlow Cytometry (BD Biosciences).

### HR activity assays for XRCC6-dependent cells

The PD(+/−)XRCC6 cells transfected with the CRISPR/Cas9 lentiviral vector were infected with lentiviruses expressing either control (pLKO5-tRFP; Addgene 57823) or sgXRCC6 (pLKO5-sgXRCC6-tRFP). After 12–16 h, the cells were transfected with Lipofectamine 2000 as recommended by the manufacturer (Invitrogen) with circular pDR-GFP (Addgene 26,475). After 24 h, the cells were serially transfected with I-SceI producing plasmid pCBASceI (Addgene 26477). After additional 24 h, the GFP-positive cells were quantified on AccuriFlow Cytometry (BD Biosciences). The average HR frequency of three experiments was measured.

The PD(+/−)XRCC6 cells infected with lentiviruses expressing either control or sgXRCC6 (including CRISPR/Cas9) were grown overnight on coverslips and then treated with γ-irradiation (2 Gy). The cells were then fixed with 4% formaldehyde, permeabilized with 0.1% Triton X-100, and blocked with 3% BSA in PBS at room temperature for 1 h. We labeled the cells with primary antibodies for Rad51 (1:500; Abcam) or Ku70 (1200, Santa Cruz), and then stained them with the corresponding Alexa Fluor 488-conjugated secondary antibody (Invitrogen, Carlsbad, CA, USA) for 1 h at room temperature. DAPI was used for nuclear staining. The cells were then mounted with Aqua-Poly-Mount mounting medium and imaged using an inverted fluorescence microscope (Carl Zeiss AG, Oberkochen, Germany). Nuclear Rad51 foci were counted within a region containing at least 50 nuclei by using local maxima in fluorescence intensity.

### EGFR nuclear localization in RAN-dependent cells

The PD(+)RAN cells were cultured in RPMI media containing 5% FBS, 1% penicillin/streptomycin, hEGF, hydrocortisone, and transferrin. Viruses for control sgRNA, sgRAN-1, or sgRAN-2 were treated three times at the intervals of 12 h. For EGFR stimulation, the cells were starved overnight with serum-free media. After hEGF treatment (100 ng/ml), the cells were harvested at 2 h and 24 h for western blotting of pEGFR and EGFR. Briefly, the cells were harvested by PBS-EDTA (5 mM) and centrifuged at 6000 rpm for 5 min. After discarding supernatant, the extraction of nuclear and cytoplasm proteins was carried out following the manufacturer's protocol (Thermo Scientific, #78833 kit). 10~50 μg of proteins were separated on SDS PAGE, transferred to a nitrocellulose membrane, and probed with antibodies for pEGFR, Cox2, and CyclinD1 (1:1000; Cell Signaling Technology). The membranes were then stripped and reprobed with antibodies for GAPDH (1:1000; Santa Cruz Biotechnology) and Lamin A/C (1:1000; Santa Cruz Biotechnology) to ensure equal loading and fractionation of nucleus and cytoplasm.

### Western blot assays

The PD(+)PSMB4 cells were harvested 48~72 h after the infection of viruses for control sgRNA, sgPSMB4-1, or sgPSMB4-2. Total protein extract was prepared. Fifteen to

30 μg of proteins were used per lane. The blot was probed with anti-Cas9 (1:1000; Santa Cruz Biotechnology, sc-517386), anti-PSMB4 (1:500; Santa Cruz Biotechnology, sc-390878), anti-Bad (1:1000; Cell Signaling Technology, ab9239), anti-Bim (1:1000; Cell Signaling Technology, ab2819), and anti-Cytochrome C (1:1000; Abcam, ab13575) antibodies. The PD(+)XRCC6 and PD(−)XRCC6 cells were harvested 1, 3, and 5 days after the infection of viruses for control sgRNA and sgXRCC6. A total of 10–20 μg of proteins was used per lane. The blot was probed with anti-DNA-PKcs (1:200; Gene-Tex), Ku70 (1:1000; Santa Cruz Biotechnology), and beta-actin (1:1000; Santa Cruz Biotechnology) antibodies. The relative densities of bands were analyzed with the NIIH ImageJ 1.47v software.

## Supplementary information

---

**Additional file 1: Figure S1.** Prediction performance by precision and recall. **Figure S2.** Prediction performance with the ARACNe network. **Figure S3.** Comparison of simulation results with experimental perturbation data in MCF7. **Figure S4.** Differential dependency of tumor versus normal samples on genes with copy number changes. **Figure S5.** Differential dependency of tumor versus normal samples on genes with point mutations. **Figure S6.** Differential expression of some alternative NHEJ genes in tumor versus normal samples. **Figure S7.** Additional Annexin V and BrdU data for RAN. **Figure S8.** Additional Annexin V and BrdU data for XRCC6. **Figure S9.** Additional Annexin V and BrdU data for PSMB4. **Figure S10.** Images of the full uncropped scans for Fig. 7a and Fig. 7b. **Figure S11.** Increases in the level of proapoptotic proteins by PSMB4 knockout. **Figure S12.** Changes in DNA-PKcs expression in response to XRCC6 knockdown.

**Additional file 2: Table S1.** Calibration parameters for deep learning.

**Additional file 3: Table S2.** Overrepresented function of predicted cancer-specific vulnerabilities in clinical samples.

**Additional file 4: Table S3.** Common cancer-specific vulnerabilities.

**Additional file 5: Table S4.** Functional enrichment of overexpressed genes in RAN-dependent samples (CRISPR+RNAi).

**Additional file 6: Table S5.** Bayesian network in breast cancer.

**Additional file 7: Table S6.** ARACNe network in breast cancer.

**Additional file 8: Table S7.** The sequences of the sgRNAs.

**Additional file 9.** Review history.

---

### Review history

The review history is available as Additional file 9.

### Peer review information

### Authors' contributions

KJ developed the model and analyzed the data. JSP, HH, and MKS assisted the data analyses. MJP and SC conducted functional experiments. SK, JJ, JWL, and S-YA assisted the functional experiments. KJ and JKC conceived the study. SC supervised the functional experiments. KJ, MJP, SC, and JKC wrote the manuscript. The authors read and approved the final manuscript.

### Funding

This research was supported by the Bio & Medical Technology Development Program of the National Research Foundation of Korea funded by the Ministry of Science and ICT (NRF-2017M3A9A7050612 and NRF-2019M3A9B6064688).

### Availability of data and materials

The codes for in silico CRISPR/RNAi and the deep learning model were made available at GitHub (http://github.com/kaistomics/DeepDependency) [51] and Zenodo (https://zenodo.org/record/3885013, DOI: https://doi.org/10.5281/zenodo.3885013) [52]. All networks used in this work (the real, shuffled, and inverted Bayesian/ARACNe networks in breast cancer and the Bayesian/ARACNe networks in liver cancer) and the TCGA prediction results are available at http://omics.kaist.ac.kr/resources. We used the ARACNe software available at http://califano.c2b2.columbia.edu/aracne [39]. We downloaded screening results for dependencies from https://depmap.org/portal/download and https://score.depmap.sanger.ac.uk/downloads. The cancer exome and transcriptome data were obtained from the TCGA database [13].

**Ethics approval and consent to participate**
Not applicable.

**Consent for publication**
Not applicable.

**Competing interests**
The authors declare that they have no competing interests.

**Author details**
[1]Department of Bio and Brain Engineering, KAIST, Daejeon 34141, Republic of Korea. [2]Department of Biomedical Sciences, University of Ulsan College of Medicine, Asan Medical Center, Seoul 05505, Republic of Korea. [3]Department of Surgery, University of Ulsan College of Medicine, Asan Medical Center, Seoul 05505, Republic of Korea. [4]Penta Medix Co., Ltd., Seongnam-si, Gyeongi-do 13449, Republic of Korea.

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

## 

