## [**Additional file 9.** Review history. · Genome Biology]

Review History

First round of review

Reviewer 1

Are you able to assess all statistics in the manuscript, including the appropriateness of statistical tests used? Yes, and I have assessed the statistics in my report.

Comments to author:

This manuscript presents a deep-learning based method to predict tumor specific vulnerabilities, especially its application in clinical samples. The predicted results, Ran, Ku70/80, were experimentally validated using patient-derived cells.

I have a few major concerns about the method, the result, and the interpretation of the results.

1. The description of methods, especially the in silico CRISPR/RNAi methodology, is confusing.
--The four equations in page 20, line 6 do not make sense to me as the sum of two conditional probabilities (e.g., $P(Y=\text{down}|X_j=\text{down})+P(Y=\text{up}|X_j=\text{up})$) cannot be added directly. Why is the probability of $P(X_j=\text{activator})$ written in this form?
--Page 19, line 21: setting up $y'=0$ for CRISPR knockout and $y'=0.2y$ for RNAi assumes that CRISPR has 100% knockout rate and RNAi has 80% knockdown rate. This is definitely not the case as even for CRISPR knockout the efficiency vary a lot between different systems. What is the rationale of setting 0.2, not 0.1, 0.3, 0.4, etc.?
-- Page 4, line 25: why specifically use liver cancer network, not other cancer types, as randomized network?
2. The validation of the methods is mainly based on AUC curve. The AUPR would be a better evaluation as the number of positive/negative points may be imbalanced (more non-dependency than dependency genes). Also, other evaluations are needed, especially validations across different datasets, including using CRISPR as training dataset and RNAi as validation dataset (and vice versa), using CRISPR datasets from different papers as separate training/validation dataset. Besides AUC/AUPR evaluation, the authors also need to confirm the model can predict known dependencies in breast cancer, for example, ER+ samples should have stronger dependency on ER; HER2+ samples should have stronger dependency on HER2 (and maybe CDK4/6); those with PIK3CA mutations should be more dependent on PIK3CA, etc.
3. The interpretation of predicting tumor but not normal dependencies is not convincing. Especially in Figure 3C, there's a large overlap of predicted tumor-specific vulnerabilities with housekeeping/core-essential genes. This indicate that many of them are important for normal cells as housekeeping genes play important fundamental biological functions. These genes will be poor therapeutic targets. Indeed, the top hit in the model, RAN, is characterized by depmap.org as "common-essential" that is vulnerable in 676/676 CRISPR screen datasets and 699/719 RNAi datasets. Not sure why it is considered as the top cancer-specific dependencies in the model. Figure 5A shows RAN dependent samples have higher RAN expression mostly in HER2 samples, but the degree of higher expression is weak and impossible to judge its statistical significance.
4. The term "predisposing" in the section "Point mutations as predisposing factors to vulnerabilities" is very confusing. If a variant is predisposing, that means it's a germline variant, not somatic mutation. The method description on page 26, line 7 is therefore confusing ("For the analysis of predisposing mutations, we used somatic variant calls for the 110 tumor that ..."). Also, the definition of TSGs and oncogenes is very loose that may contain a lot of random mutations: genes that have inactivation mutations in >5 samples as putative TSGs, and genes carrying the same substitution in >2 samples are considered as oncogenes. Why not use known TSGs/oncogenes instead?
5. The prediction of context-dependent vulnerability and synthetic lethality. As is pointed out in #3, RAN is a bad case for context-dependent vulnerability. XRCC6/PSMB4 are not tumor-specific vulnerable as well since they are both marked as "common-essential" in depmap. Furthermore, Figure 6b/c demonstrates XRCC6 and PSMB4 independent samples (PD-XRCC6-1 and PD-PSMB4-1) are indeed dependent on XRCC6/PSMB4 knockout, since they shows an increase of Annexin V and decrease of BrdU signals upon gene knockout (Figure 6B-c). The rate increasing Annexin V and decreasing BrdU for PD+Xrcc6-2 and PD-Xrcc61/2 is almost the same, indicating that they have no difference in terms of XRCC6 dependency. The only sample that shows stronger dependency to Xrcc6 is PD+Xrcc6-1, but there are confounding factors that include higher KO efficiencies for PD+Xrcc6-1 cell lines, and/or faster proliferation rate for PD+Xrcc6-1.

6. Synthetic lethality prediction. The way the authors predicts synthetic lethality is quite ad hoc: samples with stronger dependency of XRCC5/6 are predicted, based on existing biological knowledge, to be HR deficiency and are sensitive to NHEJ/DNA-PK inhibition. First, this synthetic lethal prediction is not generalizable, as it only applies to samples that is predicted to be dependent to HR/NHEJ related genes. Second, experimental validation only proves PD+XRCC6 samples are HR deficient (Fig. 7E-F). No experiments are performed on whether they are more sensitive to NHEJ/DNA-PK inhibition; the only evidence is to cite previous publications [27,28]. This is a key part to demonstrate the application of synthetic lethality.

Reviewer 2

Are you able to assess all statistics in the manuscript, including the appropriateness of statistical tests used? Yes, and I have assessed the statistics in my report.

Comments to author:

In this study, the authors have developed a deep learning-based method, which using in vitro loss-of-function screening data in cancer cell lines and exploiting cancer regulatory networks based on transcriptome data, to predict tumor-specific vulnerabilities for clinical samples with breast cancer as a model. The performance and the applicability of the prediction models were demonstrated by experimental validation for the potential dependencies of Ran, Ku70/80 and a proteasome subunit utilizing patient-derived xenografts. The strength of the method is to predict cancer-specific dependencies in clinical samples.

Overall, the manuscript is poorly written which needs a substantial revision, the manuscript is very difficult to understand, main issues remain to be addressed including:

Major

1. In general, there is a lack of explanation that why the DNN model trained on screening data in cancer cell lines could be applied to clinical samples. The two different transcriptome datasets are for tumor samples and cell lines respectively. Furthermore, evidence has pointed out that there are genomic differences between cancer cell lines and TCGA clinical samples. There exist a great gap between cell line and solid tumor sequencing data. You should able to explain the reason why the model could be "transferred" by evaluating the similarity between the expression profile of cancer cell lines and the expression profile of TCGA samples.

2. In paragraph 3 of RESULTS, the authors mentioned that "better performance can be achieved with more stringent cutoffs". In the manuscript, there were 3 cutoffs and the model achieved best performance in the most stringent cutoff. So, if the cutoff became more stringent, would the model achieve better performance or an optimal cutoff arise?

3. Please describe in detail the reason for choosing DNN over traditional machine learning models for doing subsequent analysis. There were several different metrics for model comparing and DNN was not the best in all metrics. It seems to me that the DNN is arbitrarily applied here without clear indications.

Minor

1. Please clearly cite the transcriptome data used to build the ARACNe-based regulatory network. In the manuscript, it only mentioned that the same transcriptome data used for Bayesian network construction were employed for ARACNe-based regulatory network construction. However, the transcriptome data used for Bayesian network construction did not explicitly pointed out or cited in this manuscript.

2. In paragraph 2 of RESULTS, the sentence, "Also, we randomly chose the same number of instances below the given cutoff", is not precise enough. First, only the cutoff of BAGEL can be described as "below"; the cutoffs of CERES and zGARP should be described as "above". Second, it would be easier to understand if the authors pointed out that the randomly selected instances were used as negative samples.

3. In the classification model construction section, five-fold cross-validation was used to choose an optimal hyperparameter combination for each fold, resulting in 5 different models. However, in the subsequent analysis of TCGA breast cancer data, there was a DNN model to predict the dependencies of the TCGA samples. So which dataset and hyperparameter settings were used for this DNN model training should be elaborated.

4. There are some issues should be corrected in the figures.

In Fig. 1, "Cancer specific vulnerability" should be corrected to "Cancer-specific vulnerability".

In Fig. 1, in the DNN plot, the input layer and hidden layers are not fully connected. (The top nodes in the input

layer should be connected to the bottom nodes in the hidden layers and vice versa)

In Fig. 2C, "CERES:-1.5_BAGEL:4" should be corrected to "DNN CERES:-1.5_BAGEL:4"; "Real network 0.823" and "Liver network 0.681" should be corrected to "Real network=0.823" and "Liver network=0.681".

In Fig. 3B, for heatmaps, in general, red color represents a positive correlation and blue color represents a negative correlation.

In Fig. 3C, in the third subplot the place of "RNAi" and "CRISPR" should be exchanged.

In Fig. 4D, "P = 7.04e-06" should be corrected to "P = 7.04 × 10⁻⁶".

Reviewer 1 – Overall comments

This manuscript presents a deep-learning based method to predict tumor specific vulnerabilities, especially its application in clinical samples. The predicted results, Ran, Ku70/80, were experimentally validated using patient-derived cells. I have a few major concerns about the method, the result, and the interpretation of the results.

We thank the reviewer for the time and constructive comments that we believe have considerably improved our manuscript.

Reviewer 1 – Comment 1

The description of methods, especially the in silico CRISPR/RNAi methodology, is confusing.

1) The four equations in page 20, line 6 do not make sense to me as the sum of two conditional probabilities (e.g., $P(Y=\text{down}|X_j=\text{down})+P(Y=\text{up}|X_j=\text{up})$) cannot be added directly. Why is the probability of $P(X_j=\text{activator})$ written in this form?

2) Page 19, line 21: setting up $y'=0$ for CRISPR knockout and $y'=0.2y$ for RNAi assumes that CRISPR has 100% knockout rate and RNAi has 80% knockdown rate. This is definitely not the case as even for CRISPR knockout the efficiency vary a lot between different systems. What is the rationale of setting 0.2, not 0.1, 0.3, 0.4, etc.?

3) Page 4, line 25: why specifically use liver cancer network, not other cancer types, as randomized network?

1) We thank the reviewer for pointing out this error. The equations are now corrected as follow.

$$P(X_j = \text{activator}) = \frac{P(Y = \text{up} \cap X_j = \text{up}) + P(Y = \text{down} \cap X_j = \text{down})}{P(X_j = \text{up}) + P(X_j = \text{down})}$$

$$P(Y = \text{activator}) = \frac{P(X_j = \text{up} \cap Y = \text{up}) + P(X_j = \text{down} \cap Y = \text{down})}{P(Y = \text{up}) + P(Y = \text{down})}$$

$$P(X_j = \text{inhibitor}) = \frac{P(Y = \text{down} \cap X_j = \text{up}) + P(Y = \text{up} \cap X_j = \text{down})}{P(X_j = \text{up}) + P(X_j = \text{down})}$$

$$P(Y = \text{inhibitor}) = \frac{P(X_j = \text{down} \cap Y = \text{up}) + P(X_j = \text{up} \cap Y = \text{down})}{P(Y = \text{up}) + P(Y = \text{down})}$$

2) We agree with the reviewer on the variation of the efficiency of CRISPR knockout and RNAi knockdown. Even with successful transfection of cells, the target DNA or RNA will show different suppression rates due to a multitude of factors including the nucleotide composition and G/C content [2,3]. However, it is not possible to reflect all unidentified factors and align with experimental data. Instead, we perform the simulation under ideal

suppression rates (100% and 80%, respectively) and let the deep learning model find patterns embedded in the experimental errors.

We sought to check whether this parameter affects the performance of our deep learning by running our simulation while varying the suppression rate. In previous work, CRISPR-based gene suppression efficiency ranged 90% ~ 99%, and RNA-based gene suppression 66% ~ 95% with minimal off-target effects [4,5]. Considering these reports, we randomly selected the suppression rate from a uniform distribution of 0 ~ 0.2 for CRISPR and 0.1 ~ 0.5 for RNAi. As can be seen in the figure below, the random suppression models showed similar performance as the original models.

3) The reason we chose liver cancer for negative control was that a Bayesian network in liver cancer was available from our previous work [6]. The construction of a Bayesian network is laborious and complicated, and requires a lot of resources such as chromatin interactome, TF binding, and histone modification data. In contrast, ARACNe network construction only requires gene expression data. Thus, to test whether other cancer types can serve as negative control as well, we constructed ARACNe networks in lung cancer and melanoma cancer using TCGA expression profiles. In the same way as in Fig. 2C, we built the prediction models with these two cancer networks and compared the performance levels. As shown in the figure below, the lung and melanoma models resulted in similar levels of performance as the liver model.

Reviewer 1 – Comment 2

The validation of the methods is mainly based on AUC curve. The AUPR would be a better evaluation as the number of positive/negative points may be imbalanced (more non-dependency than dependency genes). Also, other evaluations are needed, especially validations across different datasets, including using CRISPR as training dataset and RNAi as validation dataset (and vice versa), using CRISPR datasets from different papers as separate training/validation dataset. Besides AUC/AUPR evaluation, the authors also need to confirm the model can predict known dependencies in breast cancer, for example, ER+ samples should have stronger dependency on ER; HER2+ samples should have stronger dependency on HER2 (and maybe CDK4/6); those with PIK3CA mutations should be more dependent on PIK3CA, etc.

Although the ratio of the positive and negative cases was set to 1:1 for model generation, the imbalance may occur during the shuffling of data for five-fold cross validation. Fortunately, this may not be the main issue as the AUPR of our models was in good agreement with the AUC results (see the graphs below provided in Figure S1 in comparison to Fig. 2).

As suggested by the reviewer, we also performed cross-validation between the CRISPR and RNAi datasets. For this, we used cell lines and genes for which both data were available and the results were consistent. As shown below, the CRISPR-trained-RNAi-tested model and the RNAi-trained-CRISPR-tested model showed reasonable performance.

Model cross over test

Regarding the biological evaluations the reviewer suggested, we first examined whether the cell line screening data support the expected dependencies. As shown below, PIK3CA mutants were not enriched for dependencies on PIK3CA. Similarly, a tiny fraction of ER+ cell lines showed dependencies on ESR1. In contrast, 30~40 % of HER2+ cell lines turned out to be dependent on ERBB2.

*IHC, Intrinsic, and NEVE are breast cancer subtyping methods.

Therefore, we examined whether our model predicts the association between ERBB2 dependency and HER2 status. As seen below, HER2+ samples showed higher prediction scores for ERBB2. In addition, the hypergeometric test confirmed that the samples predicted to be ERBB2-dependent are enriched for the HER2+ subtype. (IHC P value = 0.0072, PAM50 P value = 0.0023).

For more detailed analyses of ERBB2 dependency, we performed further tests using results from an improved statistical framework (siMEM) that identified dependencies of HER2+ samples from screening experiment data [7]. We compared the magnitude of our prediction scores between the HER2+ and HER2- samples for the genes identified by siMEM as dependencies in the HER2+ samples. As seen below, AKT1, CDC37, CDK4, ERBB2, ERBB2IP, ERBB3, RHEB, SPDEF, and YBX1, were predicted to be more dependent in HER2+ samples than in HER2- samples by our method as well.

Reviewer 1 – Comment 3

The interpretation of predicting tumor but not normal dependencies is not convincing. Especially in Figure 3C, there's a large overlap of predicted tumor-specific vulnerabilities with housekeeping/core-essential genes. This indicates that many of them are important for normal cells as housekeeping genes play important fundamental biological functions. These genes will be poor therapeutic targets. Indeed, the top hit in the model, RAN, is characterized by depmap.org as "common-essential" that is vulnerable in 676/676 CRISPR screen datasets and 699/719 RNAi datasets. Not sure why it is considered as the top cancer-specific dependencies in the model. Figure 5A shows RAN dependent samples have higher RAN expression mostly in HER2 samples, but the degree of higher expression is weak and impossible to judge its statistical significance.

We thank the reviewer for pointing out this important issue.

First, as indicated in Fig. 3C, only 4.9% of the housekeeping genes $(81+29+78)/(81+29+78+3616)$ were predicted to be dependencies (not in all tumors but only

in particular tumors). More importantly, it has been reported that housekeeping genes can become vulnerability in a dosage-dependent manner. For example, Nijhawan et al. [1] identified cancer-specific vulnerabilities that are the result of copy-number losses and termed them CYCLOPS (Copy-number alterations Yielding Cancer Liabilities Owing to Partial losS). The CYCLOPS genes are enriched for housekeeping function such as spliceosome, proteasome, and ribosome components. They say that “Even though proteins within these pathways may be essential in all cells, genetic alterations may induce a state where reliance on these pathways creates a therapeutic window as a result of cancer-specific stresses.” and that “We identified a class of genes, enriched for cell essential genes, most predominantly proteasome, spliceosome, and ribosome components, which render cells that harbor copy-number loss highly dependent on the expression of the remaining copy.”

This should not be restricted to copy number losses. As described in detail in our manuscript, vulnerabilities will increase in cancer relative to normal when the activity of the given gene itself has been modified by not only copy number alterations but also mutations and gene expression changes (referred to as self-CNA, self-mutation, and self-DEG). In some cases, the mutations or expression changes increasing dependencies can occur in not the given gene itself but a different, functionally related gene (synthetic lethality). In other cases, cellular conditions (growth profiles, environmental stresses, genetic backgrounds, etc.) can determine the degree of dependencies on particular genes (context-dependent vulnerability).

Here we tested the above scenarios for the 188 housekeeping genes. For each gene, we split the TCGA samples into dependent and independent samples. The numbers of the self-mutation and self-CNA cases were compared between the two groups by the fisher exact test ($P < 0.05$). For the self-DEG cases, we performed the t test for the expression level of the gene between the two groups ($P < 0.05$). For context-dependent vulnerability, we performed the fisher test for the numbers of a particular subclass (ER, PR, HER2, or TN) between the two groups ($P < 0.05$). As shown below, 84% of the 188 housekeeping genes corresponded to at least one of the categories. This finding illustrates that our prediction results are in agreement with biological knowledge discovered by previous experimental reports.

RAN can be regarded as an example of context-dependent vulnerability. Our analyses in Fig. 5A suggest that these tumors are especially sensitive to RAN knockdown because of their growth profiles are particularly dependent on the function of nucleocytoplasmic transport. In rapidly dividing tumors, proteins involved in chromatin formation need to be actively synthesized and transported into the nucleus. The assembly of ribosomes and spliceosomes is reliant on nucleocytoplasmic transport as the proteins are imported into the nucleus and then exported back to the cytosol after coupling with the RNAs. Indeed, there is a previous report showing that RAN is activated by growth signaling [8]. Therefore, rapidly growing tumors should be more vulnerable to RAN silencing than slowly growing tumors and normal cells are.

This explains why the screening results suggest that RAN is common essential. We suppose that the screenings were done in the cell culture environment in which rapidly growing cells will dominate the culture population. Therefore, most of the cells in the culture will show high dependencies on RAN. This must not be the case with in vivo tumor tissues as well as normal cells.

Reviewer 1 – Comment 4

The term "predisposing" in the section "Point mutations as predisposing factors to vulnerabilities" is very confusing. If a variant is predisposing, that means it's a germline variant, not somatic mutation. The method description on page 26, line 7 is therefore confusing ("For the analysis of predisposing mutations, we used somatic variant calls for the 110 tumor that ..."). Also, the definition of TSGs and oncogenes is very loose that may contain a lot of random mutations: genes that have inactivation mutations in >5 samples as putative TSGs, and genes carrying the same substitution in >2 samples are considered as oncogenes. Why not use known TSGs/oncogenes instead?

We agree with the reviewer that the term “predisposing” could imply germline variants. This term could be replaced with “sensitizing” because these mutations sensitize the mutant cells to the activity of the gene. In any case, we have removed this term throughout the manuscript.

We did use known TSGs and oncogenes from Cancer Gene Census [9] and 20/20 rules [10] in our analysis. We just extended the list to secure a sufficient number of cases for our analyses (see the figure below). As we wrote in the methods, the known TSGs and putative TSGs were combined and referred to as TSGs. Then we used combined TSGs for LOF mutation analysis in Fig. 4C. Putative oncogenes also included known oncogenes that harbor recurrent missense mutations (Fig. 4D). TSGs and oncogenes can be cell-type specific, and some of them are not even expressed in breast cancer (see the density plot below).

More importantly, many TSGs and oncogenes did not carry recurrent mutations in the small cohort of 113 samples we could analyze in this work. For these reasons, we additionally used putative TSGs and oncogenes based on recurrent mutations occurring among the available samples.

Reviewer 1 – Comment 5

The prediction of context-dependent vulnerability and synthetic lethality. As is pointed out in #3, RAN is a bad case for context-dependent vulnerability. XRCC6/PSMB4 are not tumor-specific vulnerable as well since they are both marked as "common-essential" in depmap. Furthermore, Figure 6b/c demonstrates XRCC6 and PSMB4 independent samples (PD-XRCC6-1 and PD-PSMB4-1) are indeed dependent on XRCC6/PSMB4 knockout, since they show an increase of AnnexinV and decrease of BrdU signals upon gene knockout (Figure 6B-c). The rate increasing Annexin V and decreasing BrdU for PD+Xrcc61/2 and PD-Xrcc61/2 is almost the same, indicating that they have no difference in terms of XRCC6 dependency. The only sample that shows stronger dependency to Xrcc6 is PD+Xrcc6-1, but there are confounding factors that include higher KO efficiencies for PD+Xrcc6-1 cell lines, and/or faster proliferation rate for PD+Xrcc6-1.

Unlike RAN, XRCC6 and PSMB4 turned out to be essential in a subset (35%(248/708) and 19%(102/547)) of cell lines, respectively [11]. Similar fractions were observed in breast cancer cell lines (39%(32/82) and 25%(20/80)). As a matter of fact, dependencies are quantitative measures, and therefore we need to consider dosage effect; a low dosage of inhibitors (RNAi or CRISPR vectors) can be lethal to hypersensitive samples. As shown below, the cell line screening results are also quantitative. Many samples defined as not having dependencies (gray bars above the threshold indicated by the red line) show a

certain degree of vulnerability to knockout of the gene.

In the analysis of Figure 6, we identified samples that are 'hypersensitive' to the knockdown of RAN, XRCC6, and PSMB4. As for XRCC6, we performed functional experiments to prove that the hypersensitivity of these samples can be explained by HR deficiency (Fig. 7C, 7D, 7E, and 7F).

For the second point, the reviewer suggests that the similar rates increasing Annexin V and decreasing BrdU for PD(+)XRCC6-1/2 and PD(-)XRCC6-1/2 indicate no difference in terms of XRCC6 dependency. To address the comment, we repeated the experiments including day 0, to evaluate the difference between PD(+)XRCC6-1/2 and PD(-)XRCC6-1/2 cells in more detail. The graphs below indicate that PD(+)XRCC6-1/2 show a remarkably higher increase in cell death (by Annexin V increase) than PD(-)XRCC6-1/2 between day 0 and

day 3. Likewise, a considerable BrdU reduction was observed specifically for PD(+)XRCC6-1/2 between day 0 and day 3.

Reviewer 1 – Comment 6

Synthetic lethality prediction. The way the authors predicts synthetic lethality is quite ad hoc: samples with stronger dependency of XRCC5/6 are predicted, based on existing biological knowledge, to be HR deficiency and are sensitive to NHEJ/DNA-PK inhibition. First, this synthetic lethal prediction is not generalizable, as it only applies to samples that is predicted to be dependent to HR/NHEJ related genes. Second, experimental validation only proves PD+XRCC6 samples are HR deficient (Fig. 7E-F). No experiments are performed on whether they are more sensitive to NHEJ/DNA-PK inhibition; the only evidence is to cite previous publications [27,28]. This is a key part to demonstrate the application of synthetic lethality.

First, it is well established that HR and NHEJ are the two major pathways for double-strand break (DSB) repair. Therefore, most cancer cells would require HR and/or NHEJ activity in order to resolve DSBs generated from a high rate of DNA replication and/or DNA damage. Hence, it is acceptable to count our findings as an example of synthetic lethality occurring between HR and NHEJ. In our work, we considered the main players of HR and NHEJ (similar to BRCA1/2 and PARP1). For example, XRCC6 is the key player of NHEJ acting together with DNA-PK.

Second, we showed that the PD(+)XRCC6 samples are sensitive to the knockdown of XRCC6, the core player of the NHEJ/DNA-PK pathway (Fig. 6B). As suggested by the reviewer, we performed further experiments showing that the PD(+)XRCC6 samples are indeed sensitive to NHEJ/DNA-PK inhibition by using a known NHEJ/DNA-PK inhibitor (NU-7026). First, we found that the PD(+)XRCC6 cells are significantly more sensitive to NU-7026 as indicated by 40-60% lower IC₅₀ values (left and middle panels of the figure below). In addition, the PD(+)XRCC6 cells exhibit less active HR induction when treated with NU-7026 (right panel of the figure below), thereby suggesting that HR deficiency must be responsible for lethality caused by NHEJ/DNA-PK inhibition.

These results have been added to our paper as Figure S13. The following show how our manuscript has been revised accordingly.

Regarding Ku70/80 dependency, we first tested whether the PD(+)XRCC6 cells rely on the NHEJ pathway. When XRCC6 was inactivated, the PD(+) cells maintained the activity of DNA-PKcs whereas the PD(-) cells decreased the expression of this protein (**Figure S12**). **Indeed, the PD(+) samples were more sensitive to the DNA-PK inhibitor, NU-7026, as indicated by 40-60% lower IC50 values (Figure S13A~S13B).** This reliance on the NHEJ pathway may reflect HR deficiency. It is known that HR-defective cells are sensitive to DNA-PKcs inhibition[12,13]. To test this, we challenged the PD(+/-)XRCC6 cells with γ -irradiation to generate DSBs. XRCC6 inactivation led to an increase in the number of Rad51 foci in both the PD(+) and PD(-) cells (**Fig. 7C**). However, the degree of the increase of Rad51 foci was 3~4-fold lower in the PD(+) cells than the PD(-) cells (**Fig. 7D**). We also performed the DR-GFP/I-SceI HR assay[14] in which the fraction of GFP-positive cells indicates HR activity for DSB repair. While XRCC6 suppression led to an increase in the fraction of the GFP-positive cells (**Fig. 7E**), the magnitude of increase was 3~6-fold lower in the PD(+) cells than the PD(-) cells (**Fig. 7F**). **The similar results were obtained from the DR-GFP/I-SceI assay in response to the treatment of NU-7026 (Figure S13C).** In summary, the PD(+)XRCC6 cells displayed HR deficiencies under high DSB load, consistent with our observation with the TCGA data (**Fig. 5B**). These results illustrate how Ku70/80 silencing can render particular tumors vulnerable when the sensitivity to Ku70/80 suppression can be predicted by our computational method.

Reviewer 2 – Overall comments

In this study, the authors have developed a deep learning-based method, which using in vitro loss-of-function screening data in cancer cell lines and exploiting cancer regulatory networks based on transcriptome data, to predict tumor-specific vulnerabilities for clinical samples with breast cancer as a model. The performance and the applicability of the prediction models were demonstrated by experimental validation for the potential dependencies of Ran, Ku70/80 and a proteasome subunit utilizing patient-derived xenografts. The strength of the method is to predict cancer-specific dependencies in clinical samples. Overall, the manuscript is poorly written which needs a substantial revision, the manuscript is very difficult to understand.

We thank the reviewer for acknowledging the strength of this work. We hope that the reviewer's insightful comments have been addressed properly.

Reviewer 2 – Major comment 1

In general, there is a lack of explanation that why the DNN model trained on screening data in cancer cell lines could be applied to clinical samples. The two different transcriptome datasets are for tumor samples and cell lines respectively. Furthermore, evidence has pointed out that there are genomic differences between cancer cell lines and TCGA clinical samples. There exist a great gap between cell line and solid tumor sequencing data. You should able to explain the reason why the model could be "transferred" by evaluating the similarity between the expression profile of cancer cell lines and the expression profile of TCGA samples.

We thank the reviewer for raising this point. According to the reviewer's suggestion, we examined the correlation between the expression patterns of cell lines and TCGA samples. To this end, the average expression level of each gene in breast cancer was computed across the screened cell lines and across the TCGA samples. The correlation of gene expression levels between the cell lines and TCGA samples was $R = 0.71$ (see the plots below). We used patient-derived cells (PDCs) for functional validation. As described in the manuscript, the transcriptome data were obtained from their cognate patient-derived xenograft (PDX) mice. In general, PDX samples are considered to reflect the characteristics of clinical samples whereas PDCs are thought to be equivalent to cell lines. As shown below, the transcriptomes of the PDX tissues showed high correlations with both the cell lines and TCGA samples. Most importantly, the predictions made based on the PDX transcriptome data were functionally validated for XRCC6, RAN, and PSMB4 by using PDCs. In other words, our model built based on cell line data was applied for PDX (tissue-like) expression profiles, and the prediction results were confirmed by functional experiments by using PDC (cell line-like) samples, thereby highlighting the interoperability of our model.

To further confirm that our model is capable of capturing transcriptional perturbation leading to cell death from either cell line or clinical tissue data, we compared the prediction results of dependencies in the TCGA samples and PDX samples, and the experimental results from cell lines in terms of the molecular functions enriched for the identified genes. As shown below, the gene set enrichment analyses show a very high concordance in the most significant biological terms and pathways between the TCGA, PDX, and cell line predictions.

Function	Adjusted p value					Resource
	Predicted dependency in clinical sample			Dependency in celllines		
	TCGA matched	TCGA 1000	PDX	CRISPR screens	RNAi screens	
regulation of cellular amine metabolic process (GO:0033238)	8.98E-11	9.11E-14	8.54E-15	2.86E-20	7.95E-15	GO_Biological Process
regulation of mRNA stability (GO:0043488)	6.29E-10	9.31E-13	8.66E-13	2.02E-22	7.66E-13	GO_Biological Process
regulation of G2/M transition of mitotic cell cycle (GO:0010389)	2.84E-07	1.02E-09	1.21E-11	1.82E-36	2.43E-16	GO_Biological Process
regulation of transcription from RNA polymerase II promoter in response to hypoxia (GO:0061418)	2.45E-09	1.16E-13	1.99E-14	5.09E-19	6.81E-14	GO_Biological Process
regulation of primary metabolic process (GO:0080090)	1.06E-06	3.16E-08	2.69E-08	6.03E-10	2.75E-10	GO_Biological Process
RNA binding (GO:0003723)	1.34E-06	4.94E-13	2.72E-15	1.46E-188	1.09E-75	GO_Molecular Function
Proteasome	3.10E-12	1.91E-15	1.09E-17	1.19E-22	5.59E-16	KEGG
Protein processing in endoplasmic reticulum	7.84E-08	3.45E-07	1.83E-06	0.248696196	1	KEGG
Oxidative phosphorylation	0.001402992	2.78E-04	1.11E-06	0.002846592	1	KEGG

“TCGA matched” indicates the 113 samples where matched normal transcriptome data were available whereas “TCGA 1000” indicates 1,000 tumor samples.

Reviewer 2 – Major comment 2

In paragraph 3 of RESULTS, the authors mentioned that "better performance can be achieved with more stringent cutoffs". In the manuscript, there were 3 cutoffs and the model achieved best performance in the most stringent cutoff. So, if the cutoff became more stringent, would the model achieve better performance or an optimal cutoff arise?

As shown below, we built additional models for varying cutoffs and evaluated their performance to show more systematically that better performance can be achieved with more stringent data. However, what concerns us is that more stringent data lead to a reduction in the number of training cases and therefore to an increase in the probability of overfitting to the given cases and failing to cover various dependency cases. For the shRNA screens, zGARP threshold at -4 seemed to be optimal considering the tradeoffs between the performance level and number of training cases. For the CRISPR screens, the last threshold in the plot below, corresponding to CERES:-1.5_BAGEL:4, seemed to be the border line, beyond which there is no sufficient number of true cases.

Reviewer 2 – Major comment 3

Please describe in detail the reason for choosing DNN over traditional machine learning models for doing subsequent analysis. There were several different metrics for model comparing and DNN was not the best in all metrics. It seems to me that the DNN is arbitrarily applied here without clear indications.

We agree with the reviewer that additional description is needed as to why DNN was chosen over other machine learnings. According to the prediction results, all models except Naïve Bayes Classifier (NBC) show reasonable performance (Fig. 2B). Deep Neural Network (DNN) is also successful (Fig. 2A). However, our statements regarding the superiority of the DNN model were not simply based on the performance levels.

First, the predicted patterns of tumor-specific increases in dependencies were most clearly pronounced with DNN, as indicated by the differences between the orange versus blue curves of Fig. 4A. This has been described in the manuscript as follows;

In general, tumor samples presented high variability with biases toward high levels of dependencies (Fig. 4A). As discussed in the next section, these results imply that tumor undergoes extensive somatic changes that render the cells vulnerable to the loss of certain molecular activities. Among the nonlinear methods, DNN and radial SVM exhibited this pattern most clearly (Fig. 4A). RF showed this pattern only with the CRISPR model (Fig. 4A).

[Figure 4]

Second, as described in the “Point mutations increase tumor-specific vulnerabilities” section, some cancer-specific vulnerabilities should be attributed to somatic DNA lesions that are only present in tumor cells, and this phenomenon can be used to test the functional validity of our prediction models. The differences between the blue/green versus gray curves of Fig. 4B and Fig. 4C, and the differences between the orange versus gray curves of Fig. 4D, indicate how much this phenomenon can be supported by the DNN model. In comparison, less clear distinctions were observed for the other models as shown in Figure S5.

[Figure 4]

[Figure S5]

SVM RBF

RF

NBC

SVM linear

LDA

To further address the reviewer's concern, we compared different classifiers in their prediction on HER2 dependencies of HER2+ samples. In other words, we tested how well HER2+ samples are predicted to be dependent on the ERBB2 gene by different classifiers. As shown below, the non-linear classifiers did a better job of predicting HER2 dependencies in general. In particular, DNN was slightly better than the other non-linear classifiers.

Reviewer 2 – Minor comment 1

Please clearly cite the transcriptome data used to build the ARACNe-based regulatory network. In the manuscript, it only mentioned that the same transcriptome data used for Bayesian network construction were employed for ARACNe-based regulatory network construction. However, the transcriptome data used for Bayesian network construction did not explicitly pointed out or cited in this manuscript.

We revised the description for the transcriptome data as below.

Regulatory network construction

For regulatory network construction, we **downloaded TCGA RNA-seq data for 1,215 breast cancer and 423 liver cancer samples from the UCSC Cancer browser (<https://xenabrowser.net>)**. We employed our previous Bayesian network in breast cancer[15] and the same type of network in liver cancer[16] for a negative control. We constructed another type of regulatory network on the basis of ARACNe (Algorithm for the Reconstruction of Accurate Cellular Networks)[17]. ~~The same breast and liver-cancer transcriptome data used for Bayesian network construction were employed.~~

Reviewer 2 – Minor comment 2

In paragraph 2 of RESULTS, the sentence, "Also, we randomly chose the same number of instances below the given cutoff", is not precise enough. First, only the cutoff of BAGEL can be described as "below"; the cutoffs of CERES and zGARP should be described as "above". Second, it would be easier to understand if the authors pointed out that the randomly selected instances were used as negative samples.

We thank the reviewer for pointing out the error. We have revised the manuscript as follow.

We randomly chose the same number of instances below (for BAGEL) or above (for CERES and zGARP) the given cutoffs as independency cases.

Reviewer 2 – Minor comment 3

In the classification model construction section, five-fold cross-validation was used to choose an optimal hyperparameter combination for each fold, resulting in 5 different models. However, in the subsequent analysis of TCGA breast cancer data, there was a DNN model to predict the dependencies of the TCGA samples. So which dataset and hyperparameter settings were used for this DNN model training should be elaborated.

First, we trained one hyperparameter set for five-fold cross-validation. Then we measured the average performance from five different models. The optimal hyperparameter set was chosen by the best average performance score from the five trained models. For predicting from the TCGA samples, we used these five models to get the average prediction score.

Reviewer 2 – Minor comment 4

There are some issues should be corrected in the figures.

In Fig. 1, "Cancer specific vulnerability" should be corrected to "Cancer-specific vulnerability".

In Fig. 1, in the DNN plot, the input layer and hidden layers are not fully connected. (The top nodes in the input layer should be connected to the bottom nodes in the hidden layers and vice versa)

In Fig. 2C, "CERES:-1.5_BAGEL:4" should be corrected to "DNN CERES:-1.5_BAGEL:4"; "Real network 0.823" and "Liver network 0.681" should be corrected to "Real network=0.823" and "Liver network=0.681".

In Fig. 3B, for heatmaps, in general, red color represents a positive correlation and blue color represents a negative correlation.

In Fig. 3C, in the third subplot the place of "RNAi" and "CRISPR" should be exchanged.

In Fig. 4D, "P = 7.04e-06" should be corrected to "P = 7.04x 10^{-6} ".

We thank the reviewer for pointing out the errors. We have corrected all the errors.

REFERENCE

1. Nijhawan D, Zack TI, Ren Y, Strickland MR, Lamothe R, Schumacher SE, et al. Cancer vulnerabilities unveiled by genomic loss. *Cell*. 2012;
2. Doench JG, Hartenian E, Graham DB, Tothova Z, Hegde M, Smith I, et al. Rational design of highly active sgRNAs for CRISPR-Cas9-mediated gene inactivation. *Nat Biotechnol*. 2014;
3. Holen T. Positional effects of short interfering RNAs targeting the human coagulation trigger Tissue Factor. *Nucleic Acids Res*. 2002;
4. Gilbert LA, Horlbeck MA, Adamson B, Villalta JE, Chen Y, Whitehead EH, et al. Genome-Scale CRISPR-Mediated Control of Gene Repression and Activation. *Cell*. 2014;
5. Semizarov D, Frost L, Sarthy A, Kroeger P, Halbert DN, Fesik SW. Specificity of short interfering RNA determined through gene expression signatures. *Proc Natl Acad Sci U S A*. 2003;
6. Jang K, Kim K, Cho A, Lee I, Choi JK. Network perturbation by recurrent regulatory variants in cancer. *PLoS Comput Biol*. 2017;13.
7. Marcotte R, Sayad A, Brown KR, Sanchez-Garcia F, Reimand J, Haider M, et al. Functional Genomic Landscape of Human Breast Cancer Drivers, Vulnerabilities, and Resistance. *Cell*. 2016;164:293–309.
8. Ly TK, Wang J, Pereira R, Rojas KS, Peng X, Feng Q, et al. Activation of the ran GTPase is subject to growth factor regulation and can give rise to cellular transformation. *J Biol Chem*. 2010;
9. Futreal PA, Coin L, Marshall M, Down T, Hubbard T, Wooster R, et al. A census of human cancer genes. *Nat Rev Cancer*. 2004;4:177–83.
10. Vogelstein B, Papadopoulos N, Velculescu VE, Zhou S, Diaz LA, Kinzler KW. Cancer genome landscapes. *Science* (80-) [Internet]. 2013 [cited 2014 Mar 19];340:1546–58. Available from: <http://www.pubmedcentral.nih.gov/articlerender.fcgi?artid=3749880&tool=pmcentrez&render type=abstract>
11. McFarland JM, Ho Z V., Kugener G, Dempster JM, Montgomery PG, Bryan JG, et al. Improved estimation of cancer dependencies from large-scale RNAi screens using model-based normalization and data integration. *Nat Commun*. 2018;9:4610.
12. Dietlein F, Thelen L, Reinhardt HC. Cancer-specific defects in DNA repair pathways as targets for personalized therapeutic approaches. *Trends Genet*. 2014.
13. Dietlein F, Thelen L, Jokic M, Jachimowicz RD, Ivan L, Knittel G, et al. A functional cancer genomics screen identifies a druggable synthetic lethal interaction between MSH3 and PRKDC. *Cancer Discov*. 2014;
14. Pierce AJ, Johnson RD, Thompson LH, Jasin M. XRCC3 promotes homology-directed repair of DNA damage in mammalian cells. *Genes Dev*. 1999;13:2633–8.
15. Kim K, Yang W, Lee KS, Bang H, Jang K, Kim SC, et al. Global transcription network incorporating distal regulator binding reveals selective cooperation of cancer drivers and risk genes. *Nucleic Acids Res* [Internet]. 2015;43:5716–29. Available from: <http://nar.oxfordjournals.org/lookup/doi/10.1093/nar/gkv532>

16. Jang K, Kim K, Cho A, Lee I, Choi JK. Network perturbation by recurrent regulatory variants in cancer. Kann MG, editor. PLOS Comput Biol. Public Library of Science; 2017;13:e1005449.

17. Basso K, Margolin AA, Stolovitzky G, Klein U, Dalla-Favera R, Califano A. Reverse engineering of regulatory networks in human B cells. Nat Genet. 2005;37:382–90.

Second round of review

Reviewer 1

The revisions addressed my previous comments well. A few suggestions:

1. Figure 6, right panels should incorporate the updated Annexin V and BrdU levels in the response letter.
2. Response to my previous comment 5. From the author's analysis XRCC6/PSMB4 appears to be essential in a fraction of cell lines (25%), in contrast to their roles as common essential genes in depmap.org (e.g., XRCC6 is essential in 769/769 CRISPR screening datasets). It would be good to explain the difference between the author's analysis and the analysis in depmap.org.

Reviewer 2

Although most of my questions were answered by the authors, there are some additional points that needed to be addressed.

1. In the answer to Major comment 2, zGARP threshold at -5 rather than -4 seemed to be optimal considering the tradeoffs between the performance level and number of training cases. First, as the figure shows, the model performance improved significantly in the process of threshold from -2 to -5 and the curve of AUC became smooth after the threshold at -5. Second, as far as the number of dependencies is concerned, the number of dependencies at threshold -5 is more consistent with the number of dependencies in the CRISPR datasets. In consequence, zGARP threshold at -5 is better than -4.
2. In the answer to Major comment 3, RF can achieve similar performances as DNN in several validation tests except the last one, the prediction on HER2 dependencies of HER2+ samples, which may have contingencies. In addition to performance, RF also has better interpretability than DNN. So, the authors can try to exploit RF model and may get some findings based on the importance score of the RF model.
3. Please explicitly point out the reasons why you finally chose DNN rather than other machine learning models in the manuscript. It will be difficult for readers to understand if you do not clearly explain the reasons.
4. Please conduct Kolmogorov–Smirnov test for the distribution of the dependency scores between tumor and matched normal, and attach P values to Fig. 4A since you compared the distributions in the manuscript.

Reviewer 1 – Comment 1

Figure 6, right panels should incorporate the updated Annexin V and BrdU levels in the response letter.

Figure 6B has now been updated.

Reviewer 1 – Comment 2

Response to my previous comment 5. From the author's analysis XRCC6/PSMB4 appears to be essential in a fraction of cell lines (25%), in contrast to their roles as common essential genes in depmap.org (e.g., XRCC6 is essential in 769/769 CRISPR screening datasets). It would be good to explain the difference between the author's analysis and the analysis in depmap.org.

Because our prediction model is based on the scoring scheme of 'gene effect scores' (CERES, DEMETER2, etc.), we examined the essentiality of XRCC6/PSMB4 in cell lines using these scores from the CRISPR/RNAi screening data. The webpage (depmap.org) uses the 'probability of dependency' that ranges 0 ~ 1. Both the gene effect scoring and our prediction results deny that these genes are common essentials. In other words, our prediction results are consistent with the scoring scheme for screening experiments that our model is based on.

Reviewer 2 – Comment 1

In the answer to Major comment 2, zGARP threshold at -5 rather than -4 seemed to be optimal considering the tradeoffs between the performance level and number of training cases. First, as the figure shows, the model performance improved significantly in the process of threshold from -2 to -5 and the curve of AUC became smooth after the threshold at -5. Second, as far as the number of dependencies is concerned, the number of dependencies at threshold -5 is more consistent with the number of dependencies in the CRISPR datasets. In consequence, zGARP threshold at -5 is better than -4.

In previous screening results, the average number of dependencies per cell line ranged from a few hundred up to one thousand [1,2]. It was more important for us to cover such variable dependency cases rather than lower the bar to be consistent with the CRISPR model, which was based on a smaller number of cell lines than the RNAi model (25 or 28 cell lines for CRISPR versus 77 lines for RNAi). The -4 model also exceeded the usual AUC cutoff of 0.8. Most importantly, we have proved the validity of the -4 threshold model in a wide range of methods.

Reviewer 2 – Comment 2

In the answer to Major comment 3, RF can achieve similar performances as DNN in several validation tests except the last one, the prediction on HER2 dependencies of HER2+ samples, which may have contingencies. In addition to performance, RF also has better interpretability than DNN. So, the authors can try to exploit RF model and may get some findings based on the importance score of the RF model.

The RF model not only showed lower performance in the prediction of ERBB2 dependencies in HER2+ samples but also had a critical flaw in terms of the tumor-specific increases in dependencies. As shown below (Fig. 4A), the RNAi RF model failed to show increased dependencies in tumor. On the contrary, there were many cases where the RF predicted decreased dependencies in tumor.

Reviewer 2 – Comment 3

Please explicitly point out the reasons why you finally chose DNN rather than other machine learning models in the manuscript. It will be difficult for readers to understand if you do not clearly explain the reasons.

The reasons why we chose DNN, described in detail in our previous response to the reviewer's comments, are clarified in the manuscript as below.

Page 6, Line 6:

We then examined how our predictions for tumor samples differed from those for normal samples. The overall distribution of the dependency scores from different classifiers was compared between tumor and matched normal. In general, tumor samples presented high variability with biases toward high levels of dependencies (**Fig. 4A**). As discussed in the next section, these results imply that tumor undergoes extensive somatic changes that render the cells vulnerable to the loss of certain molecular activities. **Among the nonlinear methods, DNN and radial SVM exhibited this pattern most clearly (Fig. 4A). RF showed this pattern only with the CRISPR model (Fig. 4A).**

Page 7, Line 13:

We tested the other classifiers in this respect. The nonlinear classifiers that made similar predictions as DNN, namely, radial SVM and RF, also showed acquired dependencies associated with the LOF or GOF mutations (**Additional file 1: Figure S5**). The only linear classifier producing results consistent with the nonlinear classifiers, NBC (**Fig. 3B**), was able to reproduce this pattern as well (**Additional file 1: Figure S5**). **However, DNN showed most clear discrepancies between the mutants and wild-type samples.**

Reviewer 2 – Comment 4

Please conduct Kolmogorov–Smirnov test for the distribution of the dependency scores between tumor and matched normal, and attach P values to Fig. 4A since you compared the distributions in the manuscript.

We have conducted KS test and revised the Figure 4A with P values as follows.

REFERENCES

1. Tsherniak A, Vazquez F, Montgomery PG, Weir BA, Kryukov G, Cowley GS, et al. Defining a Cancer Dependency Map. *Cell*. 2017;170:564-576.e16.
2. Behan FM, Iorio F, Picco G, Gonçalves E, Beaver CM, Migliardi G, et al. Prioritization of cancer therapeutic targets using CRISPR–Cas9 screens. *Nature*. 2019;